# SKAP2 is required for defense against *K. pneumoniae* infection and neutrophil respiratory burst

Giang T Nguyen[1], Lamyaa Shaban[2], Matthias Mack[3], Kenneth D Swanson[4], Stephen C Bunnell[1,5], David B Sykes[6], Joan Mecsas[1,2,7]*

[1]Graduate Program in Immunology, Tufts Graduate School of Biomedical Sciences, Boston, United States; [2]Graduate Program in Molecular Microbiology, Tufts Graduate School of Biomedical Sciences, Boston, United States; [3]Department of Internal Medicine II, University Hospital Regensburg, Regensburg, Germany; [4]Brain Tumor Center and Neuro-Oncology Unit, Department of Neurology, Harvard Medical School, Beth Israel Deaconess Medical Center, Boston, United States; [5]Department of Immunology, School of Medicine, Tufts University, Boston, United States; [6]Center for Regenerative Medicine, Massachusetts General Hospital, Boston, United States; [7]Department of Molecular Biology and Microbiology, School of Medicine, Tufts University, Boston, United States

**Abstract** *Klebsiella pneumoniae* is a respiratory, blood, liver, and bladder pathogen of significant clinical concern. We show that the adaptor protein, SKAP2, is required for protection against *K. pneumoniae* (ATCC 43816) pulmonary infections. *Skap2-/-* mice had 100-fold higher bacterial burden when compared to wild-type and burden was controlled by SKAP2 expression in innate immune cells. *Skap2-/-* neutrophils and monocytes were present in infected lungs, and the neutrophils degranulated normally in response to *K. pneumoniae* infection in mice; however, *K. pneumoniae*-stimulated reactive oxygen species (ROS) production in vitro was abolished. *K. pneumoniae*-induced neutrophil ROS response required the activity of SFKs, Syk, Btk, PLCγ2, and PKC. The loss of SKAP2 significantly hindered the *K. pneumoniae*-induced phosphorylation of SFKs, Syk, and Pyk2 implicating SKAP2 as proximal to their activation in pathogen-signaling pathways. In conclusion, SKAP2-dependent signaling in neutrophils is essential for *K. pneumoniae*-activated ROS production and for promoting bacterial clearance during infection.

*For correspondence:
joan.mecsas@tufts.edu

**Competing interests:** The authors declare that no competing interests exist.

## Introduction

*Klebsiella pneumoniae* is an opportunistic Gram-negative pathogen that can cause a wide range of life-threatening infections, including pneumonia, sepsis and urinary tract infections (*Bengoechea and Sa Pessoa, 2019*; *Paczosa and Mecsas, 2016*; *European Centre for Disease Prevention and Control, 2018*; *Weiner et al., 2016*). *K. pneumoniae* is a leading cause of hospital-associated infections with one recent study reporting that *K. pneumoniae* contributes to 7.7% of cases from over 4500 hospitals (*Weiner et al., 2016*). The increasing number of severe, and sometimes systemic, *K. pneumoniae* infections are largely attributed to a rise in antibiotic-resistant strains (*Falagas et al., 2014*; *European Centre for Disease Prevention and Control, 2018*; *Kobayashi et al., 2016*) and hyper-virulent strains that generate thicker capsules (*Zhang et al., 2016*; *Harada et al., 2019*; *Lam et al., 2018*). Highlighting the importance of the innate immune response, neutropenic patients are highly susceptible to life-threatening respiratory and bloodstream infections, including those caused by *K. pneumoniae* (*Zhu et al., 2018*; *Micozzi et al., 2017*). *K. pneumoniae* lung pathogenesis has been

**eLife digest** *Klebsiella pneumoniae* is a type of bacteria that can cause life-threatening infections – including pneumonia, blood stream infections, and urinary tract infections – in hospitalized patients. These infections can be difficult to treat because some *K. pneumoniae* are resistant to antibiotics. The bacteria are normally found in the human intestine, and they do not usually cause infections in healthy people. This implies that healthy people's immune systems are better able to fend off *K. pneumoniae* infections; learning how could help scientists develop new ways to treat or prevent infections in hospitalized patients.

In healthy people, a type of immune cell called neutrophils are the first line of defense against bacterial infections. Several different proteins are needed to activate neutrophils, including a protein called SKAP2. But the role of this protein in fighting *K. pneumoniae* infections is not clear.

To find out what role SKAP2 plays in the defense against pneumonia caused by *K. pneumoniae*, Nguyen et al. compared infections in mice with and without the protein. Mice lacking SKAP2 in their white blood cells had more bacteria in their lungs than normal mice. The experiments showed that neutrophils from mice with SKAP2 produce a burst of chemicals called "reactive oxygen species", which can kill bacteria. But neutrophils without the protein do not. Without SKAP2, several proteins that help produce reactive oxygen species do not work.

Understanding the role of SKAP2 in fighting infections may help scientists better understand the immune system. This could help clinicians to treat conditions that cause it to be hyperactive or ineffective. More studies are needed to determine if SKAP2 works the same way in human neutrophils and if it works against all types of *K. pneumoniae*. If it does, then scientists might be able use this information to develop therapies that help the immune system fight infections.

extensively investigated in a mouse model using the rodent-adapted ATCC 43816 (*Bengoechea and Sa Pessoa, 2019*; *Lawlor et al., 2005*; *Xiong et al., 2015*; *Bachman et al., 2015*; *Vornhagen et al., 2019*; *Batra et al., 2012*). In murine models of infection, the *K. pneumoniae* 43816 strain robustly infects lungs eliciting a strong innate immune response through rapid and concurrent recruitment of neutrophils and iMOs to the infected lungs (*Lawlor et al., 2005*; *Xiong et al., 2015*; *Cai et al., 2010*; *Batra et al., 2012*; *Sharma et al., 2014*; *Ye et al., 2001*). Work using this strain has revealed several genes that protect *K. pneumoniae* against the host neutrophil response (*Paczosa et al., 2020*; *Silver et al., 2019*).

Neutrophils are the first responder cell type for fighting against invading pathogens, however, their activation is tightly regulated to prevent severe tissue damage that have been correlated with several autoimmune and inflammatory diseases (*Mócsai, 2013*). At the site of infection, neutrophils can bind to pathogens leading to the activation of various killing mechanisms, including phagocytosis, generation of reactive oxygen species (ROS), degranulation, and release of neutrophil extracellular traps (NETs) (*Futosi et al., 2013*; *Nguyen et al., 2017*; *Mócsai, 2013*). Although the *K. pneumoniae* capsule decreases bacterial binding and internalization (*Regueiro et al., 2006*; *March et al., 2013*), ROS, degranulation, NETs, and cytokine production have all been implicated in host defense against *K. pneumoniae* (*Bengoechea and Sa Pessoa, 2019*; *Chen et al., 2016*; *Hirche et al., 2005*; *Ivin et al., 2017*; *Paczosa et al., 2020*; *Papayannopoulos et al., 2010*; *Zhao et al., 2015*). The importance of ROS in host protection is highlighted in the setting of Chronic Granulomatous Disease (CGD), a disease characterized by genetic mutations in the NADPH oxidase, where patients are unable to make ROS and are susceptible to *Klebsiella* infections (*Wolach et al., 2017*; *Bortoletto et al., 2015*). Furthermore, mice with defective ROS production (*Cybb-/-*) have a higher bacterial burden than wild-type mice during *K. pneumoniae* lung infection (*Paczosa et al., 2020*).

Receptor-mediated ROS production in neutrophils is driven by signal-transduction pathways that activate components of the NADPH oxidase complex (*Nguyen et al., 2017*; *Futosi et al., 2013*). These signaling pathways are tightly regulated to prevent inappropriate activation and subsequent tissue damage (*Futosi et al., 2013*). Work in mice suggests that signaling through integrin, mincle receptors, G-protein couple receptors (GPCRs) and toll-like receptors (TLRs) are critical for protection against *K. pneumoniae* infection (*Teng et al., 2016*; *Sharma et al., 2014*; *Sharma et al., 2017*;

*Regueiro et al., 2006*; *Galvão et al., 2018*; *Bengoechea and Sa Pessoa, 2019*). In fact, multiple receptors may sense and/or bind to *K. pneumoniae* triggering ROS production. Prior studies have indicated a critical role for Src Kinase Associated Phosphoprotein-2, SKAP2 (SKAP-Hom and SKAP-55R) downstream of integrin, and GPCRs in neutrophils and macrophages (*Boras et al., 2017*; *Alenghat et al., 2012*; *Swanson et al., 2008*; *Tanaka et al., 2016*; *Mócsai et al., 2002*; *Togni et al., 2005*). SKAP2, a cytosolic adaptor protein, is required for integrin-mediated ROS production, adhesion, and migration to sterile inflammatory sites through its interaction with mediators of actin rearrangement (*Alenghat et al., 2012*; *Boras et al., 2017*; *Shimamura et al., 2013*; *Tanaka et al., 2016*). SKAP2 is also a target for inactivation by bacterial virulence factors, suggesting that it plays a key role in host defenses against infections (*Black et al., 2000*; *Rolán et al., 2013*). In humans, the *Skap2* gene has been identified as a risk locus for Type 1 Diabetes, and Crohn's disease (*Jostins et al., 2012*; *Barrett et al., 2009*). In resting cells, SKAP2 is thought to exist in an autoinhibitory conformation as a homodimer that is constitutively bound to PRAM-1 in neutrophils (*Moog-Lutz et al., 2001*; *Rolán et al., 2013*), or its homolog, ADAP, in other immune cells (*Liu et al., 1998*; *Marie-Cardine et al., 1998*; *Black et al., 2000*; *Ophir et al., 2013*). Upon receptor activation, SKAP2 changes to an active conformation that is localized to the plasma membrane, which allows it to interact other effector proteins including Src Family Kinases (SFKs) (*Bureau et al., 2018*; *Marie-Cardine et al., 1998*; *Liu et al., 1998*). In addition, activated SKAP proteins can relay information from receptors to focal adhesion kinases (FAK1/Pyk2) to other cytoskeletal mediators (*Ménasché et al., 2007*; *Boras et al., 2017*; *Tanaka et al., 2016*; *Raab et al., 2017*; *Ophir et al., 2013*).

Using a murine model of *K. pneumoniae* 43816 infection, we characterized the role of SKAP2 in pulmonary host defense against *K. pneumoniae* in this study. We demonstrate that SKAP2-expressing neutrophils contributed to the pulmonary containment of *K. pneumoniae* in lungs independent of inflammatory monocytes. Using murine neutrophils, we show that *K. pneumoniae*-induced ROS production requires SKAP2, SFKs, Syk, PLCγ, Btk, and PKC, and that SKAP2 is required for maximal *K. pneumoniae*-activated phosphorylation of the tyrosine kinases SFKs, Syk, and Pyk2.

## Results

### *Skap2-/-* mice are highly susceptible to pneumonic *K. pneumoniae* infection

To examine the role of SKAP2 in the pathogenesis of pneumonic *K. pneumoniae* infection, we intranasally infected wild-type (WT) BALB/c or BALB/c *Skap2-/-* mice with a streptomycin-resistant derivative of *K. pneumoniae* 43816, a hyper-capsule producer that belongs to one of the most common hypervirulent serotype, K2 (*Paczosa and Mecsas, 2016*; *Paczosa et al., 2020*; *Silver et al., 2019*). Previous studies using *K. pneumoniae* 43816 or a derivative of this strain have reported a 50% lethal dose at 48 hr after infection with $3-5 \times 10^3$ colony forming units (cfu) in BALB/c mice, and a robust infection with high bacterial burden and cellular infiltration in the lungs by 24 hr with mice developing disseminated infection by 48 hr in C57BL/6J (*Lawlor et al., 2005*; *Xiong et al., 2015*; *Ye et al., 2001*; *Feldman et al., 2019*). WT and *Skap2-/-* mice were infected with $5 \times 10^3$ cfu and were monitored for weight loss, bacterial colonization, and innate cell recruitment at 8, 16, and 24 hr post infection (hpi) (*Figure 1A–C*; *Figure 1—figure supplement 1A,E–J*). Consistent with prior studies, by 24 hr, infected WT mice lost 2% of their body weight, and *K. pneumoniae* reached on average $10^6$ cfu with a 5-fold increase in neutrophil presence in the infected lungs (*Figure 1A–C*; *Figure 1—figure supplement 1A*; *Xiong et al., 2015*). However, *Skap2-/-* mice lost significantly more body weight than the WT cohort at 24 hpi (*Figure 1A*) and had significantly higher bacterial loads in their lungs at 16 and 24 hpi (*Figure 1B*), indicating that *Skap2-/-* mice are less competent at controlling *K. pneumoniae* infection. The increase in bacterial burden did not appear to be due to differences in the numbers of neutrophils (CD11b$^+$ Ly6G$^{hi}$) because neutrophils were detected at comparable levels in WT and *Skap2-/-* lungs, bone marrow as well as bronchoalveolar lavage of infected mice (*Figure 1C*; *Figure 1—figure supplement 1A–D*). In addition, there was no difference in alveolar macrophages (CD11b$^{int}$ CD11c$^{hi}$), resident monocytes (CD11b$^+$ Gr1$^{lo}$), or dendritic cells (CD11b$^{hi}$ CD11c$^{hi}$) in the lungs of *K. pneumoniae*-infected WT and *Skap2-/-* mice (*Figure 1C*; *Figure 1—figure supplement 1E–J*). Histological examination showed extensive leukocyte infiltration in both the WT

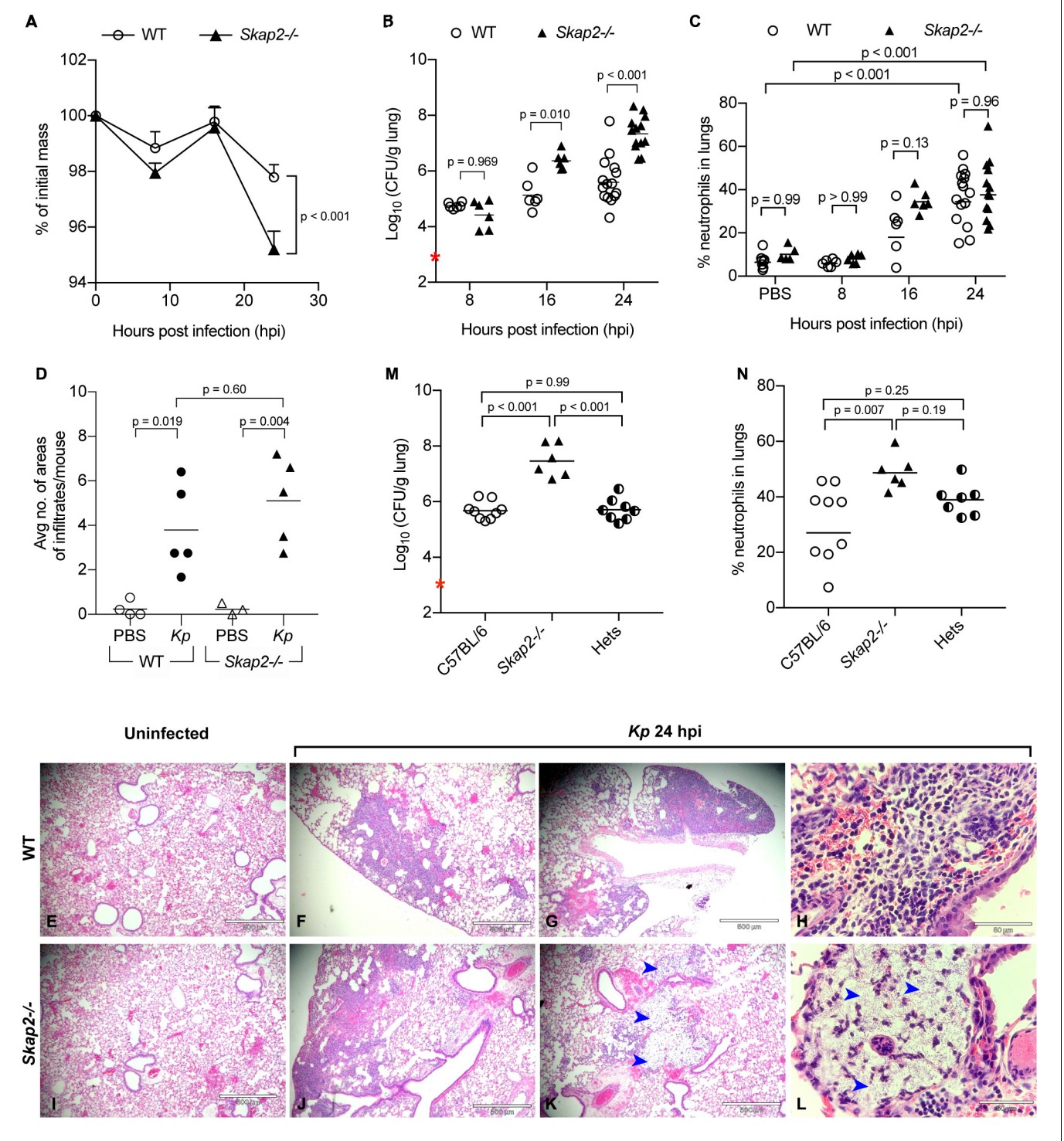

**Figure 1.** *Skap2-/-* mice are more susceptible to *K. pneumoniae* intranasal infection. (**A–L**) WT (BALB/c) and *Skap2-/-* mice were intranasally treated with PBS or infected with 5 × 10³ cfu (red asterisk) of *K. pneumoniae (Kp)*. At the indicated time points, (**A**) mice were weighed, lungs were harvested and single cell suspensions were prepared for (**B**) CFU and (**C**) analysis of neutrophils (CD11b⁺Ly6Gʰⁱ). (**D–L**) At 24 hr post-infection or inoculation with PBS (mock), lungs were harvested and processed for HE-staining. (**D**) Lung tissue sections were scored for infiltrates of leukocytes or bacteria. Mock (**E, I**) or *K. pneumoniae*-infected (**F–H, J–L**) WT (**E–H**), and *Skap2-/-* (**I–L**), lungs were imaged at 4X (**E–G, I–K**) or at 40X (**H, L**). Blue arrows indicate bacteria. Bacterial burden (**M**) and live neutrophils (CD11b⁺Ly6Gʰⁱ) (**N**) from *K. pneumoniae*-infected C57BL/6, *Skap2-/-*, and *Skap2+/-* (Hets) littermates are shown. Data are compiled from 2 to 4 independent experiments with 2–4 mice/time point/genotype. (**A**) Mean ± SEM. (**B–D, M–N**) Each dot represents

*Figure 1 continued on next page*

*Figure 1 continued*

values from a mouse, and black bars represent geometric means (**B, M**) or means (**C–D, N**). Significance was assessed using (**A**) two-way ANOVA with Sidak's post-test, or one-way ANOVA with Tukey's post-test (**M–N**), or with Sidak's post-test (**B–C, D**); log-transformed numbers were used for (**B, M**). The online version of this article includes the following figure supplement(s) for figure 1:

**Figure supplement 1.** WT and *Skap2-/-* mice have similar numbers of immune cells in lungs after *K. pneumoniae* intranasal infection.

and *Skap2-/-* infected lungs supporting the flow cytometry data (*Figure 1D–L*, black arrows). This interstitial infiltration inflammatory cells can be seen in the early stages of infection prior to broncho-pneumonia (*Lawlor et al., 2005*; *Xiong et al., 2015*). Furthermore, some infected *Skap2-/-* lungs (2/ 5 lungs) had lesions with cellular destruction and uncontrolled bacterial growth surrounded by sparse leukocyte infiltration (*Figure 1K–L*, blue arrows). To evaluate whether SKAP2-dependent control of *K. pneumoniae* colonization was required in a different mouse background, C57BL/6J, C57BL/6J *Skap2-/-*, or heterozygous littermates were inoculated retropharyngeally with *K. pneumoniae* (*Figure 1M–N*; *Figure 1—figure supplement 1K–N*). At 24 hpi, C57BL/6 *Skap2-/-* mice had a significant increase in bacterial burden compared to C57BL/6 and heterozygous littermates (*Figure 1M*). In general, comparable numbers of neutrophils and inflammatory monocytes were detected in all three strains (*Figure 1N*; *Figure 1—figure supplement 1L–N*) although the percent of neutrophils recovered in C57Bl/6 *Skap2-/-* mice was higher (*Figure 1N*) In summary, although leukocytes are present in the *Skap2-/-* lungs at comparable or slightly higher level to wild-type lungs, *Skap2-/-* mice were more susceptible to *K. pneumoniae* infection suggesting that their antimicrobial functions are compromised.

### *Skap2* expression in bone marrow-derived immune cells is required to control *K. pneumoniae* lung infection

To investigate whether *Skap2* expression in bone marrow-derived leukocytes was required for controlling the antibacterial response to *K. pneumoniae* infection, we generated bone marrow (BM) chimeric mice by transplanting WT or *Skap2-/-* BM cells into either WT or *Skap2-/-* mice (*Figure 2A*). Transplantation efficiency was confirmed by flow cytometry (*Figure 2—figure supplement 1*). WT recipient mice reconstituted with *Skap2-/-* BM had significantly more *K. pneumoniae* in contrast to WT mice transplanted with WT BM (*Figure 2B*). Conversely, *Skap2-/-* recipients with WT BM cells had significantly decreased *K. pneumoniae* in their lungs as compared to *Skap2-/-* hosts that received *Skap2-/-* BM cells (*Figure 2B*). The differences in bacterial burdens were not due to differences in numbers of neutrophils as equivalent numbers were recovered among all four groups of mice (*Figure 2C*; *Figure 2—figure supplement 1C–E*). While fewer inflammatory monocytes (iMOs) were detected in *Skap2-/-* recipients regardless of BM cell donor, the difference did not correlate with bacterial load (*Figure 2B,D*; *Figure 2—figure supplement 1F–H*). Combined, these data indicate *Skap2* expression in BM-derived immune cells is required to contain *K. pneumoniae* growth during lung infection, and that *Skap2* expression in cells that are not derived from the marrow do not contribute to this phenotype.

### *Skap2* expression in neutrophils contributes to the protection against *K. pneumoniae* infection

The contribution of neutrophils and iMOs to host defense against *K. pneumoniae* infection is dependent on the bacterial strain used with *K. pneumoniae* ATCC 43816 being more susceptible to neutrophils (*Xiong et al., 2015*; *Silver et al., 2019*). To determine the role of neutrophils and iMOs in the increased susceptibility of *Skap2-/-* mice to *K. pneumoniae*, WT and *Skap2-/-* mice were treated with the α-Ly6G (clone 1A8) to deplete neutrophils, or the MC-21 antibody, which targets CCR2$^+$ cells to deplete iMOs (*Figure 2E*). Depletion efficiency was confirmed by flow cytometry (*Figure 2—figure supplement 2A–F*). While the *K. pneumoniae* burden in iMOs-depleted WT mice was similar to vehicle-treated mice, significantly more bacteria were recovered from the lungs of neutrophil-depleted WT mice (*Figure 2E*) confirming the importance of neutrophils in host defense (*Ye et al., 2001*; *Xiong et al., 2015*). The neutrophil-depleted WT mice had similarly high bacterial burdens as *Skap2-/-* mice (*Figure 2E*), and the depletion of *Skap2-/-* mice of neutrophils or iMOs did not

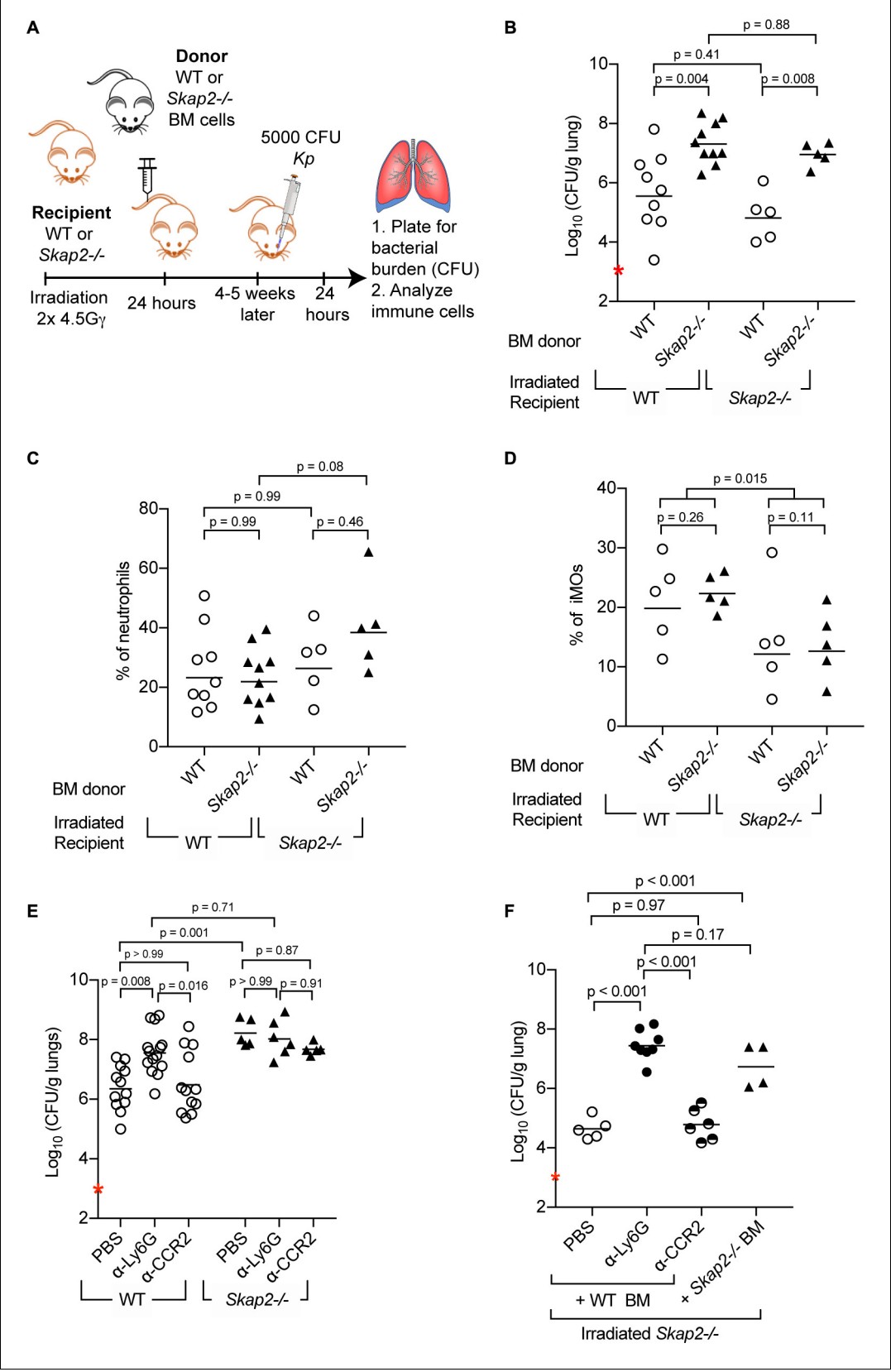

**Figure 2.** Reconstitution of *Skap2-/-* mice with WT bone marrow hematopoietic stem cells confers protection against *K. pneumoniae*. (A) Schematic used to generate bone marrow chimeras in (B–D, F). (B–F) Mice were *Figure 2 continued on next page*

*Figure 2 continued*

infected with $5 \times 10^3$ cfu *K. pneumoniae* (red asterisk); 24 hpi mice were sacrificed and lungs harvested. (**E–F**) WT and *Skap2-/-* mice were injected intraperitoneally with 50 µg of α-Ly6G (1A8) or 20 µg of α-CCR2 (MC-21) to deplete neutrophils and iMOs, respectively, or PBS 16 hr prior to infection. Bacterial burden (**B, E, F**) and percent live neutrophils (CD11b$^+$ Ly6G$^{hi}$) (**C**), or inflammatory monocytes (CD11b$^+$ Ly6C$^{hi}$) (**D**) from *K. pneumoniae*-infected lungs. Data are compiled from 2 to 4 independent experiments using groups of 2–3 mice/genotype/experiment. Each dot represents a mouse, bars are geometric means (**B, E–F**) or means (**C–D**). Statistics were assessed using one-way ANOVA with (**C–D**) Sidak's post-test, or (**B, E–F**) Tukey's post-test. (**D**) Percent of iMOs were compiled, and comparison between WT and *Skap2-/-* irradiated recipient disregarding the donor BM were assessed by two-tailed unpaired Student's *t* test.

The online version of this article includes the following figure supplement(s) for figure 2:

**Figure supplement 1.** Transplantation efficiency of bone marrow reconstitution into WT and *Skap2-/-* recipients.

**Figure supplement 2.** Evaluation of immune cell populations in depletion studies in WT and *Skap2-/-* mice.

---

increase their bacterial burden suggesting that protection against *K. pneumoniae* infection is SKAP2-dependent (*Figure 2E*).

To assess the contribution of *Skap2* expression in BM-derived iMOs and neutrophils in limiting *K. pneumoniae* infection, iMOs and neutrophils were depleted from irradiated *Skap2-/-* mice that had been reconstituted with WT BM cells (*Figure 2F*). Depletion efficiency was confirmed by flow cytometry (*Figure 2—figure supplement 2G–J*). As expected, reconstitution of *Skap2-/-* mice with WT BM resulted in a significant reduction in bacterial burden compared to *Skap2-/-* mice that received *Skap2-/-* BM cells (*Figure 2F*). *Skap2-/-* mice that received WT BM and were then depleted of neutrophils with α-Ly6G showed an increase in bacterial load and that burden was similar to those in *Skap2-/-* mice that received *Skap2-/-* neutrophils (*Figure 2F*). By contrast, α-CCR2-treated mice had a similar bacterial burden as PBS-treated mice (*Figure 2F*). Together, these data indicate that neutrophils cannot control *K. pneumoniae* 43816 infection in the absence of SKAP2 in the hematopoietic compartment. As *Skap2-/-* neutrophils are present in infected tissues (*Figure 1D,J,N*; *Figure 1—figure supplement 1A–D*), the uncontrolled *K. pneumoniae* growth may occur because of defects in the antimicrobial functions of *Skap2-/-* neutrophils at the site of infection.

## *Skap2-/-* differentiated in vitro (DIV) neutrophils phenocopied bone marrow-derived (BM) neutrophils morphologically and can used to investigate the role of SKAP2 in neutrophil functions

Neutrophils have multiple antimicrobial mechanisms to limit bacterial growth during infection (*Futosi et al., 2013*; *Mócsai, 2013*). To investigate which antimicrobial functions are controlled by SKAP2, the ER-Hoxb8 conditionally-immortalized stem cell system was adopted as a tool to produce a supply of neutrophils (*Wang et al., 2006*; *Chu et al., 2019*). Mouse BM-derived stem cells from WT and *Skap2-/-* mice were retrovirally transduced with Hoxb8 regulated by estrogen (ER-Hoxb8), and maintained in media containing estrogen (E2) and stem cell factor (SCF) to expand granulocyte-monocyte progenitors (GMP) in vitro (*Figure 3—figure supplement 1A–I* and see Materials and Methods). Removal of E2 inactivates Hoxb8 expression and the addition of interleukin-3 and granulocyte colony-stimulating factor led to terminal neutrophil differentiation (*Figure 3—figure supplement 1A–I*). Neutrophil maturation from WT and *Skap2-/-* ER-HoxB8-immortalized GMP cells (Hoxb8 GMP) was evaluated by nuclear morphology (*Figure 3—figure supplement 1B*) and cell surface expression of cKit, CD11b and Ly6G (*Figure 3—figure supplement 1C–D,G–I*). Hoxb8 GMP cells were mononuclear and cKit$^+$ CD11b$^-$ Ly6G$^-$ (*Figure 3—figure supplement 1C–D,G–I*). Cells from both mouse lines differentiated by day 4 and were comparably viable (*Figure 3—figure supplement 1E–F*). Fully differentiated cells are referred to as <u>d</u>ifferentiated <u>i</u>n <u>v</u>itro (DIV) neutrophils. Like BM-derived neutrophils, DIV neutrophils expressed polymorphonuclear morphology and are cKit$^-$ CD11b$^+$ Ly6G$^+$ (*Figure 3—figure supplement 1B–D,G–I*).

To confirm the functionality of DIV neutrophils in comparison to BM neutrophils, their ability to generate reactive oxygen species (ROS) following integrin and Fcγ receptor stimulation was assessed (*Chu et al., 2019*). When plated onto surfaces coated with poly-RGD, an engineered polypeptide containing multiple copies of the integrin binding motif, WT BM and DIV neutrophils released superoxide, while *Skap2-/-* BM and DIV neutrophils did not (*Figure 3A–C*). This is consistent with

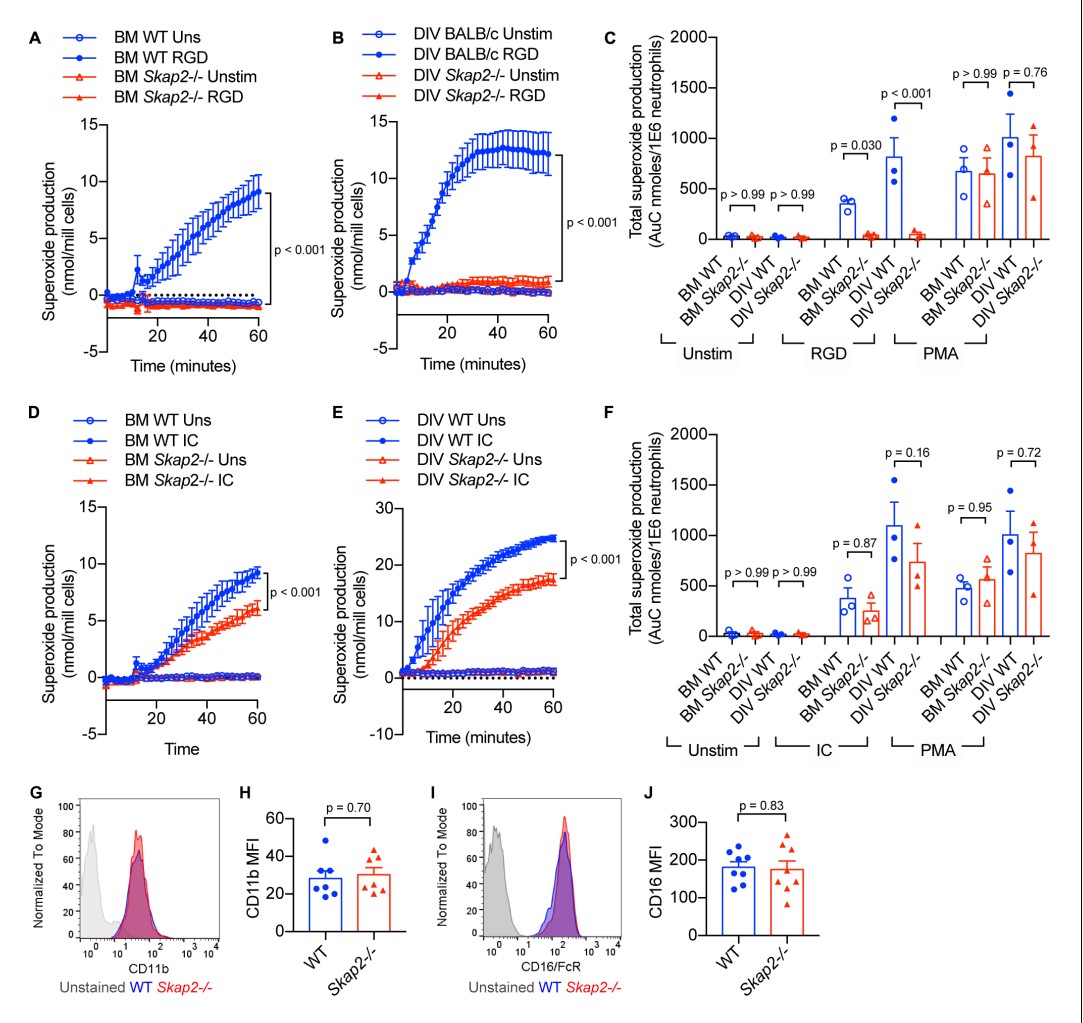

**Figure 3.** BM and DIV neutrophils require SKAP2 for integrin-activated ROS production, but not for FcγR. (A–F) Extracellular respiratory burst of BM and DIV neutrophils. WT or *Skap2-/-* neutrophils were plated on a poly-RGD-coated surface (A–C), or IgG immune complex (IC)-coated surface (D–F), and superoxide production was measured by cytochrome C reduction. Unstimulated (unstim) cells were plated onto 10% FBS/PBS or stimulated with 100 nM PMA. (C, F) Total concentration of superoxide produced after 60 min. (G–J) Expression of surface receptors on DIV neutrophils of CD11b (G–H) or activating CD16 Fcγ receptor (I–J). (G–J); Blue shaded areas and bars represent WT, and red, *Skap2-/-*. (A–B, D–E) represent the mean ± SD of one experiment in technical triplicate assessed using two-way ANOVA with Tukey's post-test. (C, F, H, J) represent the mean ± SEM of at least three independent experiments performed in at least technical duplicate (C, F) and were assessed using two-way ANOVA with Tukey's post-test (C, F), or two-tailed unpaired Student's *t* test (H, J).

The online version of this article includes the following figure supplement(s) for figure 3:

**Figure supplement 1.** Hoxb8-transformed GMP differentiate into neutrophils with similar morphology and surface markers to that of BM-derived neutrophils.

previously published data in C57BL/6J and *Skap2-/-* BM neutrophils (*Boras et al., 2017*). The loss in integrin-mediated ROS production was not due to defects in the NADPH oxidase itself, as *Skap2-/-* BM and DIV neutrophils robustly released ROS when stimulated with phorbol myristate acetate (PMA), which bypasses receptor-mediated signaling pathways and activates NADPH oxidase via protein kinase C (PKC) (*Dang et al., 2001*; *Figure 3C*). By contrast, after stimulation of Fcγ receptors by IgG immune complexes (IC), WT and *Skap2-/-* BM and DIV neutrophils robustly produced ROS (*Figure 3D–F*). *Skap2-/-* BM and DIV neutrophils consistently released 60–80% as much ROS as compared to levels seen in comparable WT neutrophils. This indicates IC-stimulated ROS is partially dependent on SKAP2. Differences in ROS production were not due to differences in surface marker expression levels as WT and *Skap2-/-* DIV neutrophils expressed similar levels of integrin and Fcγ

receptors as measured by CD11b and CD16 expression, respectively, by flow cytometry (*Figure 3G–J*). Combined, these results indicate WT and *Skap2-/-* DIVs phenocopy primary BM neutrophils both morphologically and functionally and are a useful tool to investigate how SKAP2 modulates *K. pneumoniae*-induced neutrophil responses. Furthermore, these data suggest that a functional NADPH complex can assemble in *Skap2-/-* neutrophils and that SKAP2 is required for receptor-activated signaling pathways downstream of integrin receptors, but not Fcγ receptors.

## Protection against *K. pneumoniae* is not dependent on degranulation or phagocytosis

Multiple neutrophil antimicrobial mechanisms, including degranulation, phagocytosis, and NETs, have been implicated in host defense against *K. pneumoniae* infection (*Chen et al., 2016*; *Ivin et al., 2017*; *Papayannopoulos et al., 2010*; *Bengoechea and Sa Pessoa, 2019*; *Paczosa and Mecsas, 2016*; *Hirche et al., 2005*; *Zhao et al., 2015*; *Kobayashi et al., 2016*). To examine whether *Skap2-/-* neutrophils degranulated as efficiently as WT neutrophils during *K. pneumoniae* infection, we measured levels of released MMP-9 and neutrophil elastase (ELA2), proteins found in tertiary and primary granules, respectively (*Papayannopoulos et al., 2010*; *Sheshachalam et al., 2014*). Supernatants from WT and *Skap2-/- K. pneumoniae*-infected lungs had equivalent levels of MMP-9 and ELA2 in *Skap2-/-* at 16 and 24 hpi as detected by ELISA (*Figure 4A–B*). This suggests that the increased in susceptibility of *Skap2-/-* mice was not due to defects in degranulation.

To determine the role of SKAP2 in phagocytosis of *K. pneumoniae*, phagocytosis was initially measured using microscopy. However, microscopic examination of WT DIV neutrophils infected with GFP-labeled *K. pneumoniae* and counterstained with Alexa594-conjugated polyvalent α-*K. pneumoniae* for extracellular bacteria did not reveal any internalized bacteria indicating that phagocytosis levels were less than 1% (unpublished data). Using a more sensitive gentamicin protection assay (*Walker et al., 2019*), *K. pneumoniae* 43816 was recovered on average at less than 0.01% in WT and *Skap2-/-* DIV neutrophils, while a non-encapsulated (Δ*cpsB*) *K. pneumoniae* strain was recovered at approximately 0.05% (*Figure 4C*). This low level was not due to a defect in phagocytosis of DIV neutrophils as WT and *Skap2-/-* DIV neutrophils were able to phagocytose *Yersinia pseudotuberculosis* YPIII Δ*yscNU* strain, a strain that is susceptible to phagocytosis (*Balada-Llasat and Mecsas, 2006*). WT and *Skap2-/-* DIV neutrophils phagocytosed 20% and 10% of Δ*yscNU*, respectively (*Figure 4C*), suggesting that phagocytosis of this *Y. pseudotuberculosis* strain is partially dependent on SKAP2. Although we observed low level of phagocytosis of *K. pneumoniae* 43816, the SKAP2-

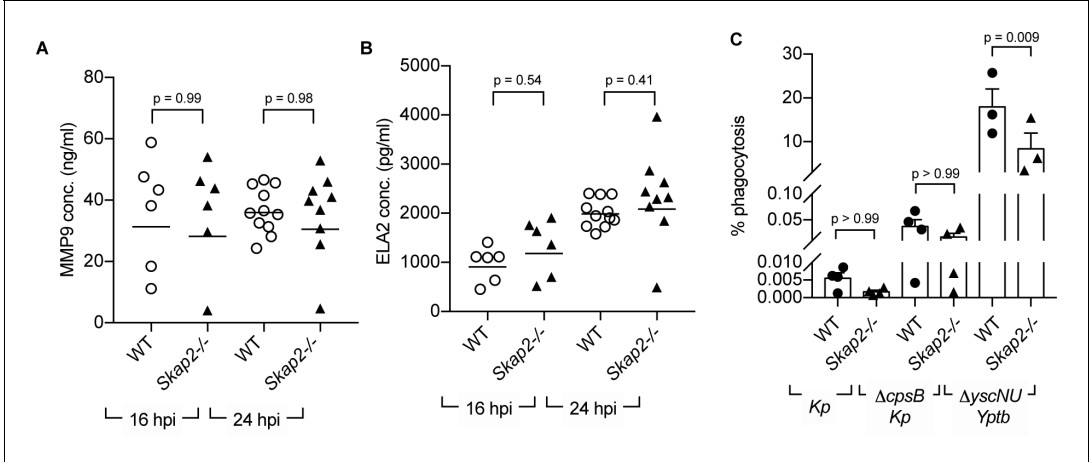

**Figure 4.** SKAP2 is not required for *K. pneumoniae*-stimulated degranulation nor phagocytosis. (**A–B**) Levels of (**A**) total MMP-9, and (**B**) neutrophil elastase (ELA2) from cell-free supernatant from *K. pneumoniae*-infected WT and *Skap2-/-* lung homogenates were analyzed by ELISA. Data are compiled from 2 to 4 independent experiments with 2–3 mice/genotype/experiment. Each dot represents a mouse and bars represent geometric means. (**C**) WT and *Skap2-/-* DIV neutrophils were incubated with encapsulated (*Kp*), unencapsulated (Δ*cpsB Kp*) *K. pneumoniae,* or *Yptb* Δ*yscNU*. Percent phagocytosis was calculated as $CFU_{bacteria\ with\ neutrophils\ and\ gentamicin}/CFU_{bacteria\ without\ gentamicin\ treatment}$. (**A–C**) Data are compiled from at least 3 independent experiments performed in technical triplicate. Statistics represent mean ± SEM and were assessed using one-way ANOVA with Sidak's post-test.

mediated defect in phagocytosis might be more prominent when challenged with other *K. pneumoniae* isolates. Combined, these results indicate that the *K. pneumoniae* 43816 strain is resistant to neutrophil phagocytosis regardless of SKAP2 expression, but can elicit SKAP2-independent degranulation during infection.

## SKAP2 is required for *K. pneumoniae*-induced neutrophil ROS production

ROS is an important mechanism for protection against *K. pneumoniae* as mice that lack the essential gp91[phox] component of NADPH oxidase, *Cybb-/-*, were significantly more susceptible to *K. pneumoniae* infection and had a significant increase in bacterial burden in their lungs (*Figure 5—figure supplement 1A*; *Paczosa et al., 2020*). To test whether SKAP2 deficiency resulted in a defect in neutrophil ROS production after *K. pneumoniae* infection, an isoluminol-amplified chemiluminiscence assay was used. In contrast to WT BM and DIV neutrophils, neither *Skap2-/-* BM nor DIV neutrophils produced ROS following exposure to *K. pneumoniae* (*Figure 5A–B*; *Figure 5—figure supplement 1B-C*). The loss of *K. pneumoniae*-stimulated ROS production was not due to a defect in bacterial-binding as WT and *Skap2-/-* DIV neutrophils bound GFP-labeled *K. pneumoniae* equivalently when assessed by flow cytometry following both 15- and 30 min incubations (*Figure 5C*). Without treating the cells with actin inhibitor, such as cytochalasin B, bound and phagocytosed bacteria could contribute to the levels of GFP[+] cells. However, the less than 1% of internalization of *K. pneumoniae* observed in the gentamicin protection assay (*Figure 4C*) suggests that the majority of the bacteria are extracellular, and that GFP[+] cells are more likely reflective of binding. Finally, SKAP2 was important for ROS production after infection with another gram-negative bacterial strain, a type 3 deficient *Y. pseudotuberculosis* strain, ΔyscF (*Figure 5—figure supplement 1D*). In summary, *Skap2-/-* neutrophils are defective in triggering the signaling cascade(s) required for ROS after *K. pneumoniae* binding to surface receptors.

To investigate how SKAP2 mediates *K. pneumoniae*-induced ROS production, we first evaluated the requirement for several well-established proteins that are critical for receptor-mediated ROS production after *K. pneumoniae* infection. Src family kinases (SFKs), Spleen Associated Tyrosine Kinase (Syk), Bruton's tyrosine kinase (Btk), Phospholipase Cγ2 (PLCγ2), and PKC play critical roles in receptor-mediated ROS production (*Löfgren et al., 1999*; *Braselmann et al., 2006*; *Volmering et al., 2016*; *Dang et al., 2001*; *Raad et al., 2009*; *Kovács et al., 2014*; *Fumagalli et al., 2007*; *Graham et al., 2007*; *Nguyen et al., 2017*). To determine whether these proteins are required for *K. pneumoniae*-stimulated ROS production, WT DIV neutrophils were pretreated with PP2, R406, Ibrutinib, U-73122, and Go6983, which are small molecules that inhibit the functions of SFKs, Syk, Btk, PLCγ2, and PKC (both novel and conventional isoforms), respectively (*Figure 5D*; *Peterman et al., 2004*; *Braselmann et al., 2006*; *Rolán et al., 2013*; *Prezzo et al., 2019*; *Fumagalli et al., 2007*). The DIV neutrophils were then infected with *K. pneumoniae*. Pretreatment with PP2, R406, Ibrutinib, U-73122, and Go6983 abolished ROS production after *K. pneumoniae* infection compared to vehicle-pretreated neutrophils (*Figure 5E*). As expected, only pretreatment with Go6983 blocked ROS after PMA exposure. U73433, an U73122 analog that does not target PLCγ, did not block *K. pneumoniae*- and PMA-induced ROS (*Figure 5E*; *Rolán et al., 2013*). This suggests SFKs, Syk, Btk, PLCγ2, and PKC are each necessary for *K. pneumoniae*-activated ROS production.

Next, we assessed whether SKAP2 mediates *K. pneumoniae*-induced ROS production by influencing tyrosine phosphorylated-dependent functions of SFKs and Syk. To determine whether SFKs and Syk are phosphorylated in response to *K. pneumoniae* infection, western blots of lysates from *K. pneumoniae*-infected WT DIV neutrophils were probed with phospho-specific antibodies. Infection with *K. pneumoniae* induced phosphorylation of SFKs (pSFK) and Syk (pSFK) to levels 2–5-fold higher than in unstimulated neutrophils (*Figure 5F–I*; *Supplementary file 1*). These results support findings with inhibitors showing that SFKs and Syk functions are required for *K. pneumoniae*-activated ROS (*Figure 5E*). In addition, proline-rich tyrosine kinase 2 (Pyk2), which is a tyrosine kinase that functions downstream of SFKs and Syk to mediate ROS production (*Kamen et al., 2011*; *Zhao and Bokoch, 2005*), was phosphorylated in response to *K. pneumoniae* stimulation (*Figure 5K–L*; *Supplementary file 1*). Thus, signaling circuit(s) containing SFKs, Syk, and Pyk2 were activated in neutrophils following receptor binding of *K. pneumoniae*. To determine if SKAP2 is required for phosphorylation of these tyrosine kinases, the levels of pSFKs, pSyk, and pPyk2 were assessed in

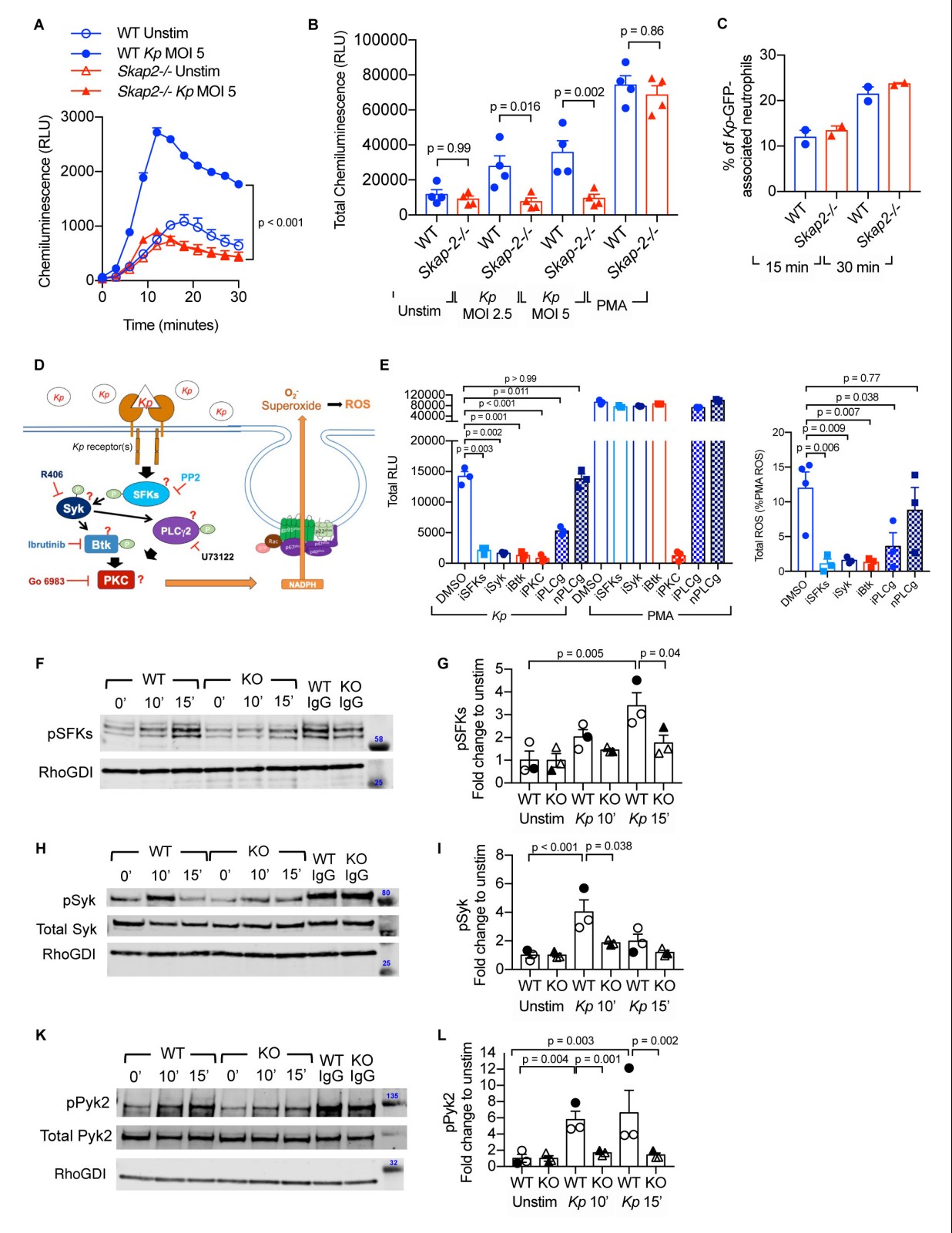

**Figure 5.** *K. pneumoniae*-stimulated ROS production requires SKAP2-dependent activation of tyrosine kinases. (A–C) Respiratory burst of WT and *Skap2-/-* DIV neutrophils infected with *Kp* using isoluminol-chemiluminsence. Unstimulated (unstim), *Kp*-infected, and 100 nM PMA-treated cells were seeded on FBS-coated wells. (A) Representative experiment performed in triplicate of ROS (RLU) production following *K. pneumoniae* stimulation. (B) Total ROS (total RLU) produced after 30 min of stimulation was calculated as the total area under the curve shown in (A) of 4 independent experiments

*Figure 5 continued on next page*

Figure 5 continued

performed in technical triplicate. (C) WT and *Skap2-/-* neutrophils incubated with GFP-expressing *K. pneumoniae* (*Kp*-GFP) at MOI 40 for 15 or 30 min, stained with DAPI, and analyzed by flow cytometry for GFP-associated neutrophils. (D) Schematic of potential *K. pneumoniae*-activated signaling pathways tested by inhibitors. (E) Respiratory burst of WT DIV neutrophils untreated or treated with inhibitors using isoluminol-chemiluminescence assay. DIV neutrophils were pre-treated with DMSO, PP2 (iSFKs), R406 (iSyk), Ibrutinib (iBtk), Go 6083 (iPKC), U73122 (iPLCγ), or U73134 (nPLCγ/non-inhibitory analog of PLCγ inhibitor) for 10 min at 37°C and then infected with MOI 2.5 of *K. pneumoniae* or treated with 100 nM PMA and measured for 30 min. Total RLU was calculated as area under the curve. Data are a representative figure from 3 independent experiments showing mean ± SD performed in technical triplicate. Significance was assessed using one-way ANOVA with Sidak's post-test. (F–L) WT and *Skap2-/-* neutrophils were infected with *Kp* for 10 or 15 min or stimulated with IC for 10 min at 37°C. Lysates were analyzed by western blot for pSFKs (Y416), pSyk (Y352), pPyk2 (Y402), and RhoGDI. Blots were then stripped and re-probed for total Syk or Pyk2. Data are compiled from 3 independent experiments. (B–C) Data are compiled from 2 to 4 independent experiments performed in technical triplicate. Statistics represent mean ± SEM. (F, H, K) Representative blot shown. (G, I, L) Solid symbols indicate values of blot shown. Bars indicate mean. Significance was assessed using one-way ANOVA (B) with Sidak's post-test between WT and *Skap-2-/-*, or (G, I, L) between time points within each genotype, or two-way ANOVA with Sidak's post-test between WT and *Skap2-/-* within the same point.

The online version of this article includes the following figure supplement(s) for figure 5:

**Figure supplement 1.** ROS restricts *K. pneumoniae* infection in lungs and is induced by *K. pneumoniae* after infection of BM neutrophils.

*Skap2-/-* DIV neutrophils following *K. pneumoniae* exposure. While SFKs, Syk, and Pyk2 phosphorylation were all slightly induced in *Skap2-/-* neutrophils, the level of induction was often significantly reduced as compared to WT neutrophils (*Figure 5F–L*; *Supplementary file 1*). To determine if the reduction in phosphorylation was due to decreased cell viability, *Skap2-/-* DIV neutrophils were also stimulated with IC as a control. Stimulation of FcγRs resulted in the phosphorylation for SFKs, Syk, and Pyk2 to similar levels in both WT and *Skap2-/-* DIV neutrophils (*Figure 5F,H,K* and *Supplementary file 1*). Collectively, these results show that SFKs, Syk, and Pyk2 are required for *K. pneumoniae*-stimulated ROS production, and that the loss of SKAP2 prevents full phosphorylation of SFKs, Syk, and Pyk2 after *K. pneumoniae* stimulation such that neutrophils are unable to generate ROS production.

## Discussion

Our experiments reveal that SKAP2 plays a key role in host defense in the hematopoietic compartment against *K. pneumoniae* 43816 in pulmonary infections in a murine lung model of infection. SKAP2 was indispensable for *K. pneumoniae*-stimulated neutrophil ROS production, but not degranulation or phagocytosis, suggesting that the increased bacterial burden in the *Skap2-/-* mice is affected by the loss of ROS. Sensitivity of *K. pneumoniae* to ROS is strongly supported by the increased bacterial burden in *Cybb-/-* mice during lung infection, and observations in CGD patients (*Wolach et al., 2017*; *Paczosa et al., 2020*). In murine neutrophils, SKAP2 was required for full phosphorylation of SFKs, Syk, and Pyk2, demonstrating a critical role in *K. pneumoniae*-induced signaling pathways that are required for ROS production. Consistent with previous findings (*Regueiro et al., 2006*; *March et al., 2013*), *K. pneumoniae* 43816 was resistant to internalization indicating that phagocytosis is most likely not the primary protective mechanism provided by neutrophils to this strain. Prior studies, including one using *K. pneumoniae* 43816, have shown that degranulation of myeloperoxidase, ELA2, and cathepsin G are important for reducing bacterial burden of *K. pneumoniae* in mice (*Hirche et al., 2005*; *Zhao et al., 2015*). However, we found that despite equivalent levels of ELA2, degranulation in *Skap2-/-* mice was not sufficient to control *K. pneumoniae* burden in the absence of ROS, indicating that neutrophil ROS and degranulation may cooperate in containing *K. pneumoniae* infection. Since degranulation occurred normally in *Skap2-/-* lungs, the pathways of ROS production are distinct from signaling pathways regulating degranulation in murine neutrophils after *K. pneumoniae* infection. Neutrophils and ROS contribute to the defense against other pathogens such as *Y. pseudotuberculosis*, *Staphylococcus aureus*, *Candida albicans*, among others (*Green et al., 2016*; *Mócsai, 2013*; *Nguyen et al., 2017*). The immunological mechanisms of SKAP2-mediated neutrophil responses to other pathogens are under investigation.

We propose the following model for SKAP2-dependent neutrophilic response to *K. pneumoniae* in mice based our findings and work of others (*Figure 6*; *Alenghat et al., 2012*; *Swanson et al., 2008*; *Togni et al., 2005*; *Timms et al., 1999*). Following *K. pneumoniae* infection, activation of

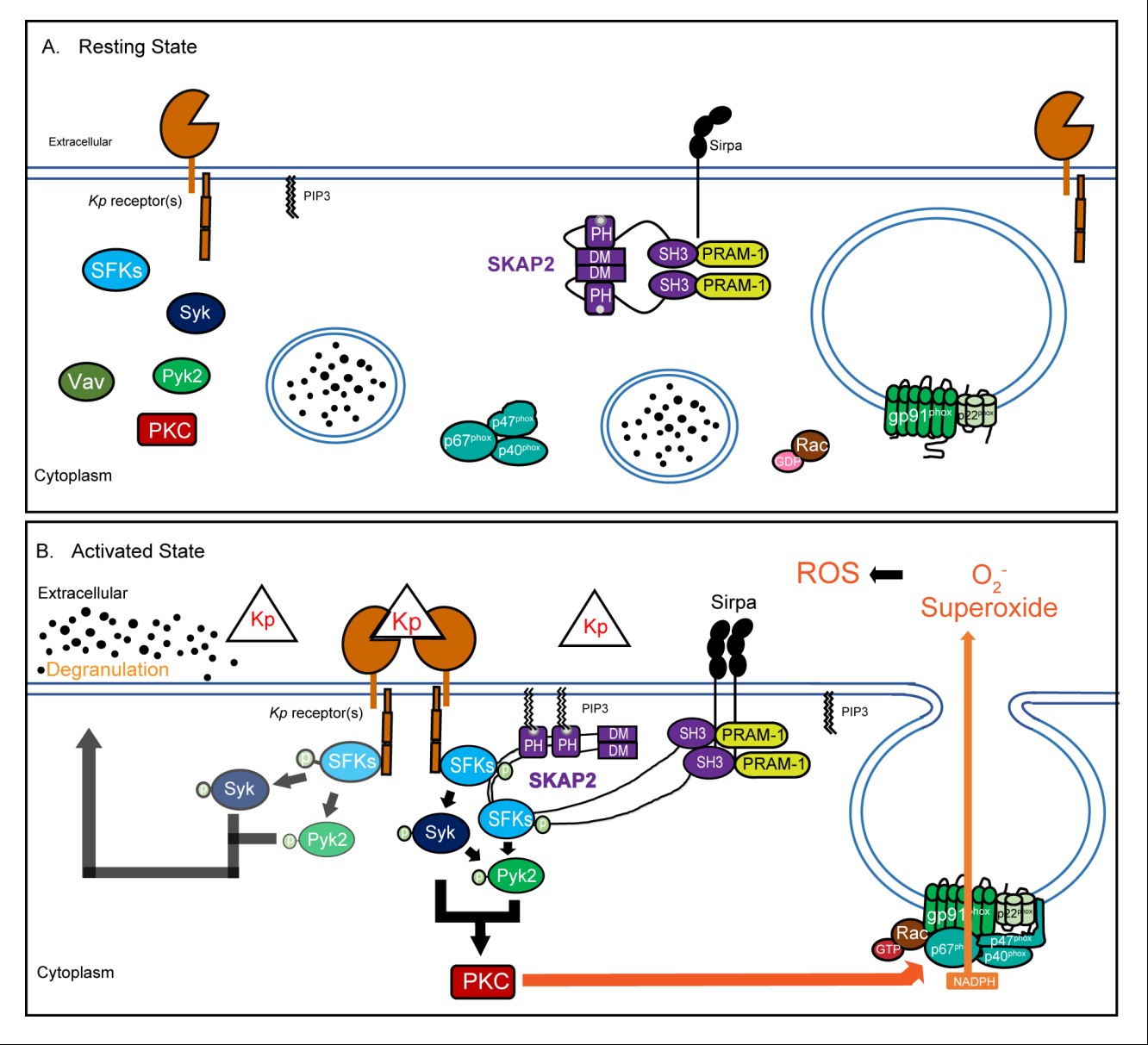

**Figure 6.** Proposed model of *K. pneumoniae*-stimulated signaling pathway. (**A**) At resting state, the homodimer SKAP2 is in an autoinhibited conformation because of binding of the DM domains; SKAP2 is constitutively associated with PRAM-1 but is apart from other components of the signaling pathways. (**B**) Activation of neutrophils through the binding of *K. pneumoniae* leads to production of PIP3 which binds SKAP2, relieving the autoinhibited conformation and revealing sites for docking and centralization of other signaling molecules. SKAP2 docking sites may centralize and retain signaling molecules thereby increasing their local concentration to facilitate increased phosphorylation and amplification of their signals. Alternatively, SKAP2 may directly activate one or more tyrosine kinases, including SFKs, Syk, and Pyk2, leading to ROS production.

receptors, like GPCRs recognizing N-formylated bacterial proteins or mincle recognizing bacterial lectins, leads to the activation of phosphoinositide 3 kinase (PI3K) and the generation of phosphatidylinositol [3,4,5]-triphosphate (PIP$_3$), resulting in PIP$_3$-mediated SKAP2 activation and membrane localization (*Figure 6A–B*; *Futosi et al., 2013*; *Swanson et al., 2008*; *Sharma et al., 2014*; *Alenghat et al., 2012*). The activated (open) phosphorylated SKAP2 complex can then bind to and enhance phosphorylation of SFKs shifting their active/inactive equilibrium to more active forms and leading to Syk activation (*Figure 6B*; *Marie-Cardine et al., 1998*; *Bureau et al., 2018*; *Kouroku et al., 1998*; *Lowell, 2011*). This is supported by previous biochemical and pull-down studies showing that SKAP2 can directly interact with SFKs (*Bureau et al., 2018*; *Liu et al., 1998*;

*Wang and Rudd, 2008*). SFKs and Syk may directly associate with and activate Pyk2, as observed in response to GPCRs, integrin, TNFα, and/or *Salmonella* stimulation (*Dikic et al., 1996*; *Evangelista et al., 2007*; *Han et al., 2003*; *Mócsai et al., 2002*). Furthermore, SKAP2 may interact with Pyk2 signaling complex directly or indirectly through their association with Sirpα (*Raab et al., 2017*; *Alenghat et al., 2012*). Any of these mechanisms would promote stabilization and propagation of SFKs/Syk/Pyk2 phosphorylation in activated murine neutrophil leading to ROS.

While SFKs, Syk, and Pyk2 have been shown to be important for degranulation in murine and human neutrophils (*Van Ziffle and Lowell, 2009*; *Kamen et al., 2011*; *Mócsai et al., 1999*; *Han et al., 2003*), we observed degranulation in murine lungs and low levels of phosphorylation after *K. pneumoniae* infection in murine neutrophils in vitro. These low levels may be sufficient to induced degranulation through a SKAP2-independent signaling pathway. Alternatively, specific phosphorylation sites of SFKs or Syk not tested here may still occurred in the absence of SKAP2 and be sufficient to induce degranulation (*Carsetti et al., 2009*; *Tsang et al., 2008*; *Moroco et al., 2014*; *Mócsai et al., 2010*). In addition, different domains of Pyk2 regulate ROS and degranulation (*Han et al., 2003*). Specifically, the inhibition of the carboxy-terminus of Pyk2 in human neutrophils leads to a reduction in ROS production in response to TNF, *Salmonella*, and *Listeria*, but did not impair neutrophil degranulation and bacterial killing (*Han et al., 2003*). SKAP2 may impact the carboxy-terminus of Pyk2 in murine neutrophils, but not activation of other domains.

*Skap2-/-* mice have normal levels of circulating neutrophils and *Skap2-/-* neutrophils were present in *K. pneumoniae*-infected lungs at a similar, and sometimes higher, levels as WT neutrophils (*Figure 1C–D*; *Figure 1—figure supplement 1A*) and (*Togni et al., 2005*). However, prior studies revealed that *Skap2-/-* myeloid cells, including neutrophils, have impaired recruitment and migration into tissues during sterile inflammation (*Tanaka et al., 2016*; *Boras et al., 2017*). This suggests that migration to sterile, but not infectious sites, may require SKAP2, which is consistent with findings with Syk. Syk is required for neutrophil migration in an inflammatory model but not during bacterial infection (*Zarbock et al., 2007*; *Frommhold et al., 2007*; *Van Ziffle and Lowell, 2009*; *Schymeinsky et al., 2006*). Alternatively, it is possible that *Skap2-/-* neutrophils may have a reduced migration level in *K. pneumoniae*-infected lungs but this is balanced by significantly higher rate of neutrophil survival and these two factors result in a net neutrophil level that is equivalent to or higher than that observed in WT mice. This result could arise because ROS production can increase myeloid cell death (*Coxon et al., 1996*; *Geering and Simon, 2011*; *Geering et al., 2011*), and thus, defects in ROS production by *Skap2-/-* neutrophils could contribute to higher survival and accumulation in infected murine lungs. These possibilities are important to address with additional experiments, such as pulse-labelling newly differentiated neutrophils to track rates of migration during infection in vivo.

Murine models of infection have been helpful in illuminating *K. pneumoniae* infection biology, however, there exist significant differences between mice and humans in the immune response to infection, including disparities in structural anatomy, circulating neutrophil counts, surface receptors, and signaling cascades, among others (*Mizgerd and Skerrett, 2008*; *Mestas and Hughes, 2004*; *Condliffe et al., 2005*; *Hajjar et al., 2010*; *Sun et al., 2016*; *Bagaitkar et al., 2012*; *Bengoechea and Sa Pessoa, 2019*). Thus, to date it is unknown if SKAP2 plays a role in mediating the human neutrophil response to *K. pneumoniae* infection. *Skap2* has 9 transcripts and at least 50 exonic variants that have been reported in the human population, however, their functional consequences, if any, remains unclear (*Cunningham et al., 2019*). It would be interesting to explore if the SKAP2-mediated *K. pneumoniae*-stimulated responses occur in human neutrophils, and whether neutrophils (or other cell types) from individuals carrying specific *Skap2* variants respond differently *K. pneumoniae* 43816 infection. In addition, it is unclear whether our results with *K. pneumoniae* 43816 will be found with other K2 strains or strains of other serotypes. Genome-wide analyses of *K. pneumoniae* clinical isolates have shown extensive genetic diversity, virulence factors, and pathogenicity (*Holt et al., 2015*; *Deleo et al., 2014*; *Agard et al., 2019*). Furthermore, since neutrophils are not the major player in host defenses against all strains of *K. pneumoniae*, including some multi-drug resistant clinical isolates (*Xiong et al., 2016*; *Xiong et al., 2015*; *Broug-Holub et al., 1997*; *Greenberger et al., 1996*), future studies can use these *K. pneumoniae* strains to probe the role of SKAP2 in neutrophils as well as other hematopoietic cells during *K. pneumoniae* infections. In summary, neutrophils and their production of ROS play a dual role in host health in many disease states from acute infections to autoimmune diseases to cancer (*Hoffmann and Griffiths, 2018*;

*Nguyen et al., 2017*), and the crucial function of SKAP2-mediated ROS in murine neutrophils suggests that further analysis of the role of SKAP2 in human health warrants investigation.

# Materials and methods

### Key resources table

| Reagent type (species) or resource | Designation | Source or reference | Identifiers | Additional information |
|---|---|---|---|---|
| Genetic reagent (*M. musculus*) | C57BL/6J | Jackson Laboratory | RRID:IMSR_JAX:000664 | |
| Genetic reagent (*M. musculus*) | B6.129S5-Skap2$^{Gt(VICTR20)21Lex}$/Mmjax | Jackson Laboratory | RRID:MGI:4353994 | |
| Genetic reagent (*M. musculus*) | B6.129S-Cybb$^{tm1Din}$/J | Jackson Laboratory | RRID:IMSR_JAX:002365 | |
| Genetic reagent (*M. musculus*) | BALB/c BALB/cAnNTac | Taconic Laboratory | RRID:IMSR_TAC:balb | |
| Genetic reagent (*Klebsiella pneumoniae*) | ATCC 43816 KPPR1 | GenBank ATCC | GenBank: CP009208.1 | |
| Antibody | Rat Anti-Ly6G monoclonal antibody, unconjugated, Clone 1A8 | Fisher Scientific | BD Biosciences Cat# 551459, RRID:AB_394206 | (100 ul of 50 ug/ml) |
| Antibody | Anti-CCR2 (MC21) | Dr. Matthias Mack *Mack et al., 2001* | Cat# CCR2 (MC21), RRID:AB_2314128 | (100 ul of 200 ug/ml) |
| Antibody | Rat anti-mouse CD16/CD32 Mouse BD Fc Block | BD Biosciences | BD Biosciences Cat# 553142, RRID:AB_394657 | (1:200 dilution) |
| Antibody | Rat monoclonal anti-mouse/human α-CD11b-PE or α-CD11b-PacBlue (clone M1/70) | Biolegend | BioLegend Cat# 101207, RRID:AB_312790 BioLegend Cat# 101223, RRID:AB_755985 | (1:300 dilution) |
| Antibody | Rat monoclonal anti-mouse α-Ly6G PE-Cy7 (clone 1A8) | Biolegend | BioLegend Cat# 127617, RRID:AB_1877262 | (1:300 dilution) |
| Antibody | Hamster monoclonal anti-mouse α-CD11c-PerCP-Cy5.5 (clone HL3) | BD Biosciences | BD Biosciences Cat# 560584, RRID:AB_1727422 | (1:300 dilution) |
| Antibody | Rat monoclonal anti-mouse α-Gr1-FITC or α-Gr1-APC (clone RB6-8C5) | BioLegend | BioLegend Cat# 108411, RRID:AB_313376 BioLegend Cat# 108405, RRID:AB_313370 | (1:300 dilution) |
| Antibody | Rat monoclonal anti-mouse α-Ly6C-AlexaFluor647 (clone HK1.4) | BioLegend | BioLegend Cat# 128010, RRID:AB_1236550 | (1:300 dilution) |

*Continued on next page*

*Continued*

| Reagent type (species) or resource | Designation | Source or reference | Identifiers | Additional information |
|---|---|---|---|---|
| Antibody | Rabbit polyclonal anti-mouse/ human α-SKAP2 | Proteintech | Proteintech Cat# 12926–1-AP, RRID:AB_2189317 | (1:200 dilution or 1.57 ug/ml) |
| Antibody | rabbit IgG polyclonal isotype antibody | Proteintech | Proteintech Cat# 30000–0-AP, RRID:AB_2819035 | (1.57 ug/ml) |
| Antibody | Alexa Fluor 488 goat anti-rabbit secondary antibody | Thermo Fisher Scientific | Thermo Fisher Scientific Cat# A-11034, RRID:AB_2576217 | (1:250 dilution) |
| Antibody | Rabbit polyclonal anti-human serum albumin | Sigma-Aldrich | Sigma-Aldrich Cat# A0433, RRID:AB_257887 | (1:400 dilution) |
| Antibody | Rabbit anti-human/mouse Phospho-Src Family (Y416) | Cell Signaling Technology | Cell Signaling Technology Cat# 2101, RRID:AB_331697 | (1:500) |
| Antibody | Rabbit anti-human/mouse monoclonal Phospho-Zap-70 (Y319)/Syk (Y352) | Cell Signaling Technology | Cell Signaling Technology Cat# 2717, RRID:AB_2218658 | (1:500) |
| Antibody | Rabbit anti-human/mouse polyclonal Phospho-Pyk2 (Y402) | Cell Signaling Technology | Cell Signaling Technology Cat# 3291, RRID:AB_2300530 | (1:500) |
| Antibody | Rabbit anti-human/mouse polyclonal RhoGDI | Cell Signaling Technology | Cell Signaling Technology Cat# 2564, RRID:AB_2274313 | (1:1000) |
| Antibody | Rabbit anti-human/mouse polyclonal Syk | Cell Signaling Technology | Cell Signaling Technology Cat# 2712, RRID:AB_2197223 | (1:500) |
| Antibody | Rabbit anti-human/mouse polyclonal Pyk2 | Cell Signaling Technology | Cell Signaling Technology Cat# 3292, RRID:AB_2174097 | (1:500) |
| Antibody | Goat Anti-rabbit IgG (H+L) (DyLight 800 4X PEG Conjugate) | Cell Signaling Technology | Cell Signaling Technology Cat# 5151, RRID:AB_10697505 | (1:20,000) |
| Recombinant DNA Reagent | MSCVneo-HA-ER-Hoxb8 | *Wang et al., 2006* | In-house (Sykes lab @ MGH) | |
| Peptide, recombinant protein | Stem cell factors | Cho-SCF | In-house | |
| Peptide, recombinant protein | Recombinant murine Stem cell factors | Peprotech | Cat. #: AF-250 | (50 ng/ml) |
| Peptide, recombinant protein | Recombinant murine Interleukin-3 | Peprotech | Cat. #: 213–13 | (10–50 ng/ml) |
| Peptide, recombinant protein | Recombinant murine Interleukin-6 | Peprotech | Cat. #: 216–16 | (10 ng/ml) |

*Continued on next page*

*Continued*

| Reagent type (species) or resource | Designation | Source or reference | Identifiers | Additional information |
|---|---|---|---|---|
| Peptide, recombinant protein | Recombinant human G-CSF | Peprotech | Cat. #: 300–23 | (50 ng/ml) |
| Peptide, recombinant protein | Fibronectin human plasma | Sigma | Cat. #: F0895 | (10 ug/ml) |
| Peptide, recombinant protein | β-estradiol | Sigma | Cat. #: E2758 | (0.5 uM) |
| Peptide, recombinant protein | Firbonectin-like protein polymer genetically engineered | Sigma Aldrich | Cat. #: F5022 | (15 ug/ml) |
| Peptide, recombinant protein | Albumin from human serum | Sigma | Cat. #: A9511 | (20 ug/ml) |
| Chemical compound | Hexadimethrine bromide (polybrene) | Sigma Aldrich | Cat. #: 107689 | (8 ug/ml) |
| Commercial assay or kit | Mouse total MMP9 DuoSet ELISA | R and D Systems | Cat. #: DY6718 | |
| Commercial assay or kit | Mouse neutrophil elastase/ELA2 DuoSet ELISA | R and D Systems | Cat. #: DY4517 | |
| Chemical compound | 4-Aminophthalhydrazide (isoluminol) | Sigma | Cat. #: A8264 | (50 uM) |
| Chemical compound | Peroxidase from horseradish | Sigma | Cat. #: P6782 | (15 U/ml) |
| Chemical compound, drug | PP2 | Selleck | Cat. #: S7008 | (10 nM) |
| Chemical compound, drug | R406 | Selleck | Cat. #: S2194 | (2 uM) |
| Chemical compound, drug | Ibrutinib (PCI-32765) | Selleck | Cat. #: S2680 | (1 uM) |
| Chemical compound, drug | Gouml 6983, PKC inhibitor | Abcam | Cat. #: ab144414 | (10 uM) |
| Chemical compound, drug | U-73122 | Sigma | Cat. #: U6756 | (1 uM) |
| Chemical compound, drug | U-73433 | Sigma | Cat. #: U6756 | (1 uM) |
| Chemical compound, drug | G418/Geneticin | Thermo Scientific | Cat. #: 10131035 | (1 mg/ml) |
| Other | Immunolon 4HBX 96-well plates | Fisher Scientific | Cat. #: 3855 | |
| Other | Cytochrome c from equine heart | Sigma | Cat. #: C7752 | (100 uM) |
| Other | Ficoll-Paque-Plus | Pharmacia/GE Healthcare | Cat. #: GE17-1440-02 | (1:1 dilution) |
| Other | Percoll | Sigma-Aldrich | Cat. #: P1644 | |

*Continued on next page*

*Continued*

| Reagent type (species) or resource | Designation | Source or reference | Identifiers | Additional information |
|---|---|---|---|---|
| Other | e-Myco Mycoplasma PCR Detection Kit | Bulldog Bio | Cat. #: 25233 | |

## Animals

BALB/c and C57BL/6J mice were purchased from Taconic Biosciences and Jackson laboratory (Bar Harbor, ME), respectively. Mating pairs of *Cybb-/-* in C57BL/6J background, and *Skap2$^{+/-}$* (B6.129S5-*Skap2$^{Gt(VICTR20)21Lex}$*/Mmjax) were purchased from Jackson laboratory and bred at Tufts University in the specific pathogen-free facility of Tufts University. Generation of *Skap2-/-* mice in the BALB/c background was previously described (*Togni et al., 2005*; *Alenghat et al., 2012*). *Skap2-/-* mice were genotyped by two independent PCRs using primer sequences: *Skap*Left-forward primer (common for both PCRs), 5' CAG CTT GCC GAC TTT TCT; GTLexVir, 5'GAG GGC TGG ACC GCA TCT GG; GT*Skap*Right, 5'CCG CCT CCC ACC CCT CAA TC following procedures described from Jackson laboratory and (*Alenghat et al., 2012*). All mice were handled in accordance with protocols approved by the Institutional Animal Care and Use Committee of Tufts University.

## Mouse intranasal infections

Mouse infections were carried out as previously described (*Paczosa et al., 2020*; *Silver et al., 2019*) with the following modifications. MKP220, streptomycin-resistant derivative of *K. pneumoniae* ATCC43816 strain was used for all infections (*Paczosa et al., 2020*; *Silver et al., 2019*). Isoflurane anesthetized mice on a BALB/c background were intranasally infected dropwise with $5 \times 10^3$ cfu *K. pneumoniae* in 40 µl of sterile PBS, or with PBS (Fisher Scientific) alone. C57BL/6J mice were inoculated retropharyngeally with $5 \times 10^3$ cfu as previously described (*Paczosa et al., 2020*). This change in infection route was because we have found that if delivered intranasally, C57BL/6J mice will rapidly dispel the inoculum through their nose and mouth resulting in inconsistent infection. Mice were euthanized using $CO_2$ asphyxiation at the indicated time points after infection. Lungs were collected into sterile PBS, weighed, and aseptically homogenized by pushing the tissue through a 70 µM cell strainer. For bacterial burden analyses, serially diluted homogenates were plated on L-agar and incubated at 37°C overnight to determine CFU/g lung. For depletion of neutrophils or iMOs, mice were intraperitoneally injecting 100 µl of 50 µg/ml α-Ly6G antibody (clone: 1A8, Fisher Scientific), or 100 µl of 200 µg/ml α-MC21 antibody 16 hr pre-infection. Depletion efficiency was measured by flow cytometry as described below. Collection of bronchoalveolar lavage was conducted as previously described (*Silver et al., 2019*).

## Bone marrow chimeras

4–5 week-old BALB/c and *Skap2-/-* mice received 100 mg/kg/24 hr SulfaTrim available ad libitum in drinking water for a week prior to and a week following whole body irradiation. Mice were lethally irradiated with 2 doses of 4.5 Gγ with a 3–4 hr interval between doses. The next day, each mouse intravenously received 100 µl of sterile PBS containing $5 \times 10^6$ BM cells from either BALB/c or *Skap2-/-* donor mice (7–15 week-old) and were infected with *K. pneumoniae* 4–5 weeks later.

## Immunohistochemistry

For histological studies, lungs from mock or *K. pneumoniae*-infected mice were perfused through a tracheal incision using 4% formaldehyde (MP Biomedicals), removed, and incubated in fresh 4% formaldehyde overnight at room temperature (RT). Excess fatty tissues were removed, and lungs were placed in histology cassettes for processing by the Tufts Animal Pathology Core. Tissues were embedded in paraffin and 7 µM thick sections were sliced and stained with hematoxylin and eosin (HE). Sections were blindly scored for number of areas of leukocyte infiltration and detectable bacteria by 4–7 individuals with at least 3 lung sections per mouse using light microscopy. Three PBS-inoculated and 5 *K. pneumoniae*-infected mice were scored per genotype. Tissues were imaged using an Echo Revolve R4 microscope in the upright brightfield configuration at 4x (NA 0.13) or 40x (NA

0.75) magnification and acquired at RT using a 12 MP CMOS color camera and the Echo Revolve built-in software.

## Degranulation

After passage of lung tissues through 70 μm strainers, the cell suspensions were centrifuged at 13,000 $x$ $g$ for 10 min at 4°C. Supernatants were then transferred into new Eppendorf tubes for measurement of release of matrix metalloproteinase (MMP-9), and neutrophil elastase (ELA-2) by DuoSet ELISA kits (R and D Systems) per manufacturer's protocols.

## Flow-cytometry analysis of lung cells

After passage of lung tissues through a 70 μm strainer, cell suspensions were treated with 1 mg/ml Collagenase D (ThermoFisher Scientific) for 1 hr at 37°C, and then with 1X Pharm Lyse (Fisher Scientific) for 5 min at 4°C. Cells were resuspended in ice-cold fluorescence-activated cell sorter (FACS) buffer (PBS containing 1% FBS), and blocked with rat anti-mouse CD16/CD32 (Mouse BD Fc Block BD Biosciences) for 10 min at 4°C. For time courses, cells were stained with α-CD11b-PE (BioLegend), α-Ly6G PE-Cy7 (BioLegend), α-CD11c-PerCP-Cy5.5 (BD Biosciences), and α-Gr1-FITC (BioLegend) for analysis of neutrophils (CD11b$^+$ Ly6G$^+$), alveolar macrophages (CD11b$^{int}$ CD11c$^{hi}$), resident monocytes (CD11b$^+$ Gr1$^{lo}$), and dendritic cells (CD11b$^{hi}$ CD11c$^{hi}$). For bone marrow chimeric and depletion experiments, lung cells were stained with α-CD11b-PacBlue (BioLegend), α-Ly6G-PE-Cy7 (BioLegend), α-Gr1-APC (BioLegend), and α-Ly6C-AlexaFluor647 (BioLegend) for analysis of neutrophils and inflammatory monocytes as indicated in figure legends. For some experiments, cells were fixed with 4% formaldehyde, permeabilized with 0.1% saponin, and intracellularly stained with α-SKAP2 (Proteintech) or rabbit IgG polyclonal isotype antibody (Proteintech), followed by goat anti-rabbit AlexaFluor488 secondary antibody (ThermoFisher Scientific). Data were collected by a BD LSRII, and analyzed with FlowJo (version 10.1).

## Bone marrow neutrophil isolation

Mouse BM neutrophils were isolated as previously described (*Rolán et al., 2013*) with the following modifications. Neutrophils were isolated using a three step Percoll density gradient (55%, 65%, and 75%), and centrifuged at 480 $x$ $g$ for 30 min at 26°C without applying the break. Neutrophils were collected at the 65–75% interface, resuspended in HBSS without Ca$^{2+}$ or Mg$^{2+}$ (HBSS-), then centrifuged at 250 $x$ $g$ for 5 min at RT to wash and pellet the cells. Supernatants were removed and cells were resuspended in HBSS- and centrifuged for a total of three washes. Neutrophils were rested at RT for one hour, resuspended in HBSS with Ca$^{2+}$ and Mg$^{2+}$ (HBSS+) to the desired concentration, incubated at RT for 20 min, and then shifted to 37°C for 10 min prior to the experiment. The purity of neutrophil preparations was verified by staining an aliquot of the cells with α-CD11b-PacBlu, α-Ly6G-PE-Cy7, and α-Gr1-APC and analyzed by flow cytometry.

## Generating HoxB8-immortalized GMP cells

Immortalization of murine BM cells with ER-HoxB8 was done as previously described (*Wang et al., 2006*) with the following modifications. Bone marrow stem cells were collected from the tibias and femurs of mice and passed through a 40 μm filter. Cells were layered over Ficoll-Paque-Plus (Pharmacia), and centrifuged at 400 $x$ $g$ for 25 min at RT without applying the break to enrich for mononuclear cells. These cells were incubated in a 6-well tissue culture plate (Greiner Bio-One) for 24 hr at 37°C with 5% CO$_2$ prior to the retroviral transduction in complete RPMI, cRPMI (RPMI-1640 with 10% FBS, 2 mM L-glutamine, and 100U penicillin and 0.1 mg/ml streptomycin), supplemented with 10 ng/ml stem cell factor (SCF), 10 ng/ml interleukin-3 (IL-3), and 10 ng/ml interleukin-6 (IL-6). The next day, non-adherent cells were harvested and $5 \times 10^5$ cells in 500 μl of cRPMI were plated onto a 12-well tissue culture plate (Corning) coated with 10 μg/ml human fibronectin (Sigma). Retroviral supernatant (MSCVneo-HA-ER-Hoxb8) was applied in the presence of 8 μg/ml polybrene and the plates were centrifuged for 'spinoculation' at 1000 $x$ $g$ for 90 min at RT. Following spin infection, cells were maintained in cRPMI supplemented with 0.5 μM beta-estradiol (Sigma, E2) and conditioned media containing approximately 50 ng/ml SCF (SCF + E2 cRPMI). Cells were maintained for 3–4 weeks to select for ER-HoxB8-carrying granulocyte-monocyte progenitor cells. Uninfected cells were maintained alongside as controls and all died by 4 weeks. To confirm the identify of cell lines,

we assessed for SKAP2 protein expression by intracellular staining or western blotting. Mycoplasma testing of cultured lines was conducted at 3–6 month intervals using e-Myco Mycoplasma PCR Detection Kit (Bulldog Bio).

## Differentiating neutrophils from Hoxb8-immortalized stem cell lines

ER-HoxB8 stem cell lines were grown in E2 + SCF cRPMI. Differentiation into neutrophils was done as previously described (*Wang et al., 2006*; *Pelletier et al., 2017*) with the following modifications. Aliquots of ER-HoxB8 progenitors were washed 3 times in PBS and cultured in E2-free cRPMI media supplemented with ~50 ng/ml SCF, IL-3, and GCSF for two days. Cells were then restimulated with 50 ng/ml GCSF for another 48 hr. At this point, cells were differentiated and termed DIV neutrophils. DIV neutrophils were collected, pelleted at 250 x *g* for 5 min at 4°C, re-suspended in HBSS- and rested for 1 hr at RT. DIV neutrophils were re-suspended in HBSS+ to the desired concentration, incubated at RT for 20 min, and then shifted to 37°C for 10 min for functional assays.

DAPI staining and flow cytometry were performed to confirm that >90% of cells were neutrophils. An aliquot of $1 \times 10^5$ BM and DIV neutrophils were spun onto microscope slide using Shannon Cyto-spin 3, fixed with 4% formaldehyde, stained with DAPI (ThermoFisher Scientific), and mounted using ProLong Gold antifade reagent (Invitrogen). Cell morphology was imaged at 60X by an Echo Revolve R4 microscope in the upright fluorescent configuration. Aliquots of cells were resuspended in cold FACS buffer and blocked with Mouse BD Fc Block (BioLegend) for 10 min at 4°C. Cells were then stained with α-CD11b-PacBlu, α-Ly6G-PE-Cy7, and α-cKit-PE, and then fixed with 4% formaldehyde prior to analysis. Viability was assessed using trypan blue (Gibco) exclusion test immediately prior to functional studies to confirm that >85% of cells were live and all counts were based on live cells. In addition, an ATP-based assay, Promega CellTiter-Glo Luminescent Cell viability Assay, was conducted during the time course of several experiments to assess viability per instructor's instructions (Promega) (**Figure 1—figure supplement 1E-F**).

## Klebsiella-neutrophil binding assay

Neutrophils ($1 \times 10^5$ cells/well) and green fluorescent protein (GFP)-labeled *K. pneumoniae* (MOI 40) were added to non-tissue-treated 96-well plates (CellTreat), centrifuged at 250 x *g* for 3 min, and then incubated for 30 min at 37°C with 5% $CO_2$ for 15 or 30 min. Cells were then harvested, centrifuged at 250 x *g* for 5 min, and resuspended in 200 ul/well ice-cold FACS buffer; this was repeated for a total of 3 washes. Cells were then fixed with 4% formaldehyde, and nuclei were stained with DAPI. The percentage of *K. pneumoniae*-associated neutrophils was determined by flow cytometry using uninfected neutrophils as control.

## Gentamicin protection assay

*K. pneumoniae* MKP220 and a isogenic non-encapsulated Δ*cpsB* strain (*Paczosa et al., 2020*; *Silver et al., 2019*) were grown overnight in LB media at 37°C with aeration, and the next day diluted 1:40 into fresh LB media and grown for an additional 2 hr at 37°C with aeration. The *Y. pseudotuberculosis* YPIII Δ*yscNU* strain (*Balada-Llasat and Mecsas, 2006*) was grown in 2XYT at 26°C with aeration. *K. pneumoniae* and *Y. pseudotuberculosis* were then used to infect $1 \times 10^5$ DIV neutrophils at an MOI of 10 for 30 min at 37°C in the presence of 5% $CO_2$ in 96-well 4HBX plates pre-coated with 10% FBS, and then 100 μg/ml gentamicin was added to wells for 1 hr at 37°C to kill extracellular bacteria. Control FBS-coated wells contained bacteria but no neutrophils and were not treated with gentamicin (non-gentamicin-treated wells). All following steps were conducted at 4°C. Cold PBS was added to the wells; plates were centrifuged at 250 x *g* for 5 min, and all media were gently removed; this was repeated for a total of four washes. All wells were treated with 0.1% Triton X-100 (Sigma-Aldrich) for 10 min at 4°C, and then serially diluted and plated to quantify surviving bacteria. The percent of phagocytosis was calculated as (# of bacteria with neutrophils from gentamicin-treated wells) / (# of bacteria from non-gentamicin-treated wells).

## Neutrophil ROS assays

High-bound tissue-treated 4HBX 96-well plates (Fisher Scientific) were used for all ROS assays. Plates were prepared as follows. For integrin activation, wells were coated with 15 μg/ml poly-RGD (Sigma) for 2 hr at 37°C or overnight at RT. For Fcγ receptor activation by IgG immune complexes (IC), wells

were coated with 20 µg/ml human serum albumin (Sigma) for 1 hr, washed twice with PBS, blocked with 10% FBS for 30 min, washed twice with PBS, and incubated with anti-human serum albumin (Sigma) at 1:400 dilution in 10% FBS for 1 hr at 37°C. Control wells were coated with 10% FBS for 30 min at 37°C. All wells were washed twice with PBS prior to use. For experiments with *K. pneumoniae*, wells were coated with 10% FBS for 30 min at 37°C and washed twice with PBS prior to use. For all ROS assays, 100 nM of PMA was used as positive control. To evaluate whether *Skap2-/-* neutrophils generated ROS after infection with another species of bacteria, a *Y. pseudotuberculosis* YPIII type 3 secretion mutant, Δ*yscF* was used (*Green et al., 2016*).

To detect RGD or IgG IC-stimulated ROS, superoxide detection by cytochrome C (Sigma) was performed as previously described (*Lowell et al., 1996*) with the following modifications. Cytochrome C was added to each well at a final concentration of 100 µM and $1 \times 10^5$ neutrophils were added to each well coated with either FBS or ligand. Absorbance at 490 nm and 550 nm were recorded at 37°C using Biotek Synergy HT. Superoxide concentration was calculated by (OD$_{550}$-OD$_{490}$) x ($2.11 \times 10^4$ M$^{-1}$cm$^{-1}$), which is the extinction coefficient of cytochrome C (*Dahlgren et al., 2007*). Total superoxide production was calculated by the sum of the area under the curves for the indicated time.

To detect ROS production following *K. pneumoniae* exposure, isoluminol was performed as previously described (*Kobayashi et al., 2016*; *Dahlgren et al., 2007*) with the following modifications. Neutrophils resuspended in HBSS+, rested at RT for 30 min, were loaded with 50 µM isoluminol (Sigma) and 15 U/ml HRP (Sigma), added to a 96-well plate at $1 \times 10^5$ cells/well, and incubated at 37°C for 10 min. *K. pneumoniae* (grown as described in gentamicin protection assays) were then added to wells at a MOI of 2.5 or 5, and the plate was centrifuged at 500 *x g* for 3 min at 4°C. For inhibitor studies, following isoluminol and HRP loading, DIV neutrophils were pre-treated with 10 nM PP2 (Selleck Chemical), 2 µM R406 (Selleck Chemical), 1 µM Ibrutinib (Selleck Chemical), 10 µM Go 6983 (Abcam), 1 µM U-73122 (Sigma), 1 µM U-73433 (Sigma), or DMSO (1:10,000 final concentration, Sigma) vehicle control for 10 min at 37°C with 5% CO$_2$. A BioTek Synergy HT late reader was used to detect chemiluminescence (RLU). Total ROS (total RLU) production was calculated by the sum of the area under the curves for the indicated time in figure legends.

## Western blot analysis

DIV neutrophils ($7.5 \times 10^5$ cells/well) in HBSS+ were plated onto 96-well 4HBX plates coated with 10% FBS. *K. pneumoniae* was added at MOI 40 and spun at 250 *x g* for 2 min. Cells were incubated at 37°C in presence of 5% CO$_2$ for 10 or 15 min, lysed in 1X Novex buffer (4X stock: 40% sucrose, 6.82% Tris-Base, 6.66% Tris-HCL, 8% SDS, 0.06% EDTA, 0.075% Bromophenol Blue, 2.5 mM NaVO$_4$, 100 mM DTT) and $5–7.5 \times 10^5$ cells equivalents were resolved on 4–12% NuPAGE gel (Invitrogen) in MOPS buffer (10.5% MOPS, 6% Tris-Base, 5% SDS, 0.3% EDTA). Proteins were transferred to Immobilon-FL PVDF (Millipore Sigma) and subjected to western blot analysis using antibodies at a dilution of 1:500 unless otherwise indicated. Antibodies to phosphorylated proteins include Src Family-Y416 (Cell Signaling Technology), Syk-Y352 (Cell Signaling Technology) and Pyk2-Y402 (Cell Signaling Technology). All blots were also probed with RhoGDI (Cell Signaling Technology) at a dilution of 1:1000 as loading control. Secondary LI-COR goat anti-mouse IRDye 800 CW (Cell Signaling Technology) and goat anti-rabbit IRDye 800 CW (Cell Signaling Technology) were used at a dilution of 1:20,000. The Odyssey CLx LI-COR system was used to develop the blots and IS Image Studio was used for analysis and quantification of bands. Blots were then stripped with 7M Guanine Hydrochloride (Fisher Scientific) by incubating at 56°C for 30 min with shaking, washed 3 times in 1x TBST (0.1% Tween-20 in 1x TBS), and reprobed with antibodies for total proteins against Syk and Pyk2 (Cell Signaling Technology). The normalized protein level of pSFKs was calculated by taking the ratio of phosphorylated protein to RhoGDI loading control. The normalized protein levels of pSyk and pPyk2 were calculated by first taking the ratio of phosphorylated to total protein and then normalizing to the RhoGDI loading control. Normalized phospho-protein levels were used to calculate the fold change to unstimulated (0 min) control within each group, or to unstimulated control of WT (*Supplementary file 1*). In addition, the relative induction of phosphorylation in unstimulated controls was determined by dividing the fold change of each experiment by the average of the three experiments. Fold change was log transformed and subjected to statistical analysis.

## Statistics

Differences between the groups were evaluated by one-way ANOVA or two-way ANOVA with either Tukey's or Sidak's post-test, where appropriate (GraphPad Software version 7). Log-transformed values were used for determining significance in experiments measuring CFU.

## Acknowledgements

This work was supported by NIH NIAID R01 AI113166 awarded to JM; LS was supported by NIH 4T32AI007422. The authors declare no competing financial interests. We thank Alyssa Fasciano, Anne McCabe, Rebecca Silver, Yoelkys Morales, Erin Green, Michelle Paczosa, Alexander Poltorak, Miles Duncan, and Maria-Cristina Seminario for critically reading the manuscript and/or for helpful scientific and technical discussions.

## Additional information

### Funding

| Funder | Grant reference number | Author |
| --- | --- | --- |
| National Institutes of Health | R01 AI113166 | Joan Mecsas |
| National Institutes of Health | 4T32AI007422 | Lamyaa Shaban |

The funders had no role in study design, data collection and interpretation, or the decision to submit the work for publication.

### Author contributions

Giang T Nguyen, Conceptualization, Formal analysis, Investigation, Visualization, Methodology, Writing - original draft, Writing - review and editing; Lamyaa Shaban, Methodology, Writing - review and editing; Matthias Mack, Kenneth D Swanson, Resources, Writing - review and editing; Stephen C Bunnell, David B Sykes, Resources, Methodology, Writing - review and editing; Joan Mecsas, Conceptualization, Supervision, Funding acquisition, Writing - review and editing

### Author ORCIDs

Giang T Nguyen (ID) https://orcid.org/0000-0002-8967-3396
Stephen C Bunnell (ID) http://orcid.org/0000-0001-6887-0828
Joan Mecsas (ID) https://orcid.org/0000-0001-9455-6672

### Ethics

Animal experimentation: All animal experimental procedures followed NIH guidelines. All mice were handled in accordance with protocols (B2018-10) approved by the Institutional Animal Care and Use Committee (IACUC) of Tufts University.

### Decision letter and Author response

Decision letter https://doi.org/10.7554/eLife.56656.sa1
Author response https://doi.org/10.7554/eLife.56656.sa2

## Additional files

### Supplementary files

• Source data 1. Figure data files.

• Supplementary file 1. Raw values of western blot analysis for phopho-Src Family Kinases, phospho-Syk, phospho-Pyk2. WT and *Skap2-/-* (KO) DIV neutrophils were infected with wild-type *Kp*, or stimulated with IgG IC (as positive control) for 10 min at 37°C. Lysates were prepared and analyzed by

western blot for phosphoproteins, total proteins, and RhoGDI as indicated. Quantification of protein level was assessed using Licor.

• Transparent reporting form

### Data availability

All data generated and analyzed during this study are included in the manuscript and supporting files.

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
