## [Decision Letter]

**Acceptance summary:**

Your investigations into the role of SKAP2 in immune defense against *Klebsiellapneumoniae*, including the elegant integration of in vivo and cell culture approaches to dissect a potential mechanistic basis for SKAP2-mediated host defense, will certainly be of interest to our readers and we are thrilled to accept it for publication.

**Decision letter after peer review:**

Thank you for submitting your article "SKAP-2 is required for neutrophil respiratory burst in response to *Klebsiella* infection" for consideration by *eLife*. Your article has been reviewed by three peer reviewers, one of whom is a member of our Board of Reviewing Editors, and the evaluation has been overseen by Jos van der Meer as the Senior Editor. The following individuals involved in review of your submission have agreed to reveal their identity: Frank DeLeo (Reviewer #1); David Rosen (Reviewer #2).

The reviewers have discussed the reviews with one another and the Reviewing Editor has drafted this decision to help you prepare a revised submission.

All three reviewers were very positive about the manuscript. The findings are interesting and important, however there are some concerns that some conclusions are overstated and more information in required in parts. In addition, context for some of the reagents and methods would be helpful for the broad readership of *eLife*. The reviewers believe that these revisions should be able to be done without additional experiments.

Major Comments:

1) The authors focus on the conclusion that they show a role for SKAP2 in neutrophils, which, based on their data is true in vitro. However, the data presented falls short of showing that this is related to the phenotypes observed in vivo. Figure 2 shows that loss of SKAP2 in hematopoetic cells is responsible for the susceptibility phenotype and that neutrophils contribute to control of this particular *Klebsiella* strain in WT mice. But the data do not show that SKAP2 has a role in neutrophils in vivo during infection, it only shows that neutrophils cannot control the infection in the absence of SKAP2 in the hematopoetic compartment. Therefore, without additional data (i.e. specific deletion in neutrophils or transferring of purified neutrophils from a knockout into a WT (or vice versa) or using a *Klebsiella* strain not reliant on neutrophils for control of infection), the authors need to change their conclusions to more accurately match the data shown throughout the manuscript.

2) The flow cytometry data should also be presented as number of cells in a population (in addition to frequency of a population). For the flow cytometry data throughout the paper, the authors show % of live cells.

3) It would be optimal if the authors had some data that shows a similar function for SKAP2 in human neutrophils. At the very least, text should be included that acknowledges the caveats (well known) of extrapolating data from mouse neutrophils to human neutrophils. The authors should be cautious in extrapolating their data to additional scenarios beyond those tested.

4) In Discussion paragraph four the authors mentioned published work that *SKAP2-/-* neutrophils exhibit a migration defect in response to sterile inflammation and imply there is no migration defect during infection. But the authors never show any data to support this implication. The authors assume that migration kinetics are the same because they measure the same number of neutrophils in the lungs during infection at specific timepoints, but they do not provide information on how many neutrophils were present in the blood and lungs of naïve mice, nor do they do a pulse labeling of newly differentiated neutrophils to track their rate of migration.

5) In this same section, the authors mention that neutrophil survival could be affected by loss of SKAP2. This is a very important point in general and in particular for when analyzing their in vitro data. Did the authors observe a difference in survival of neutrophils from SKAP2 mice? How were each of their in vitro studies normalized for total live cells at each time point?

---

## [Author Response]

Major Comments:1) The authors focus on the conclusion that they show a role for SKAP2 in neutrophils, which, based on their data is true in vitro. However, the data presented falls short of showing that this is related to the phenotypes observed in vivo. Figure 2 shows that loss of SKAP2 in hematopoetic cells is responsible for the susceptibility phenotype and that neutrophils contribute to control of this particular Klebsiella strain in WT mice. But the data do not show that SKAP2 has a role in neutrophils in vivo during infection, it only shows that neutrophils cannot control the infection in the absence of SKAP2 in the hematopoetic compartment. Therefore, without additional data (i.e. specific deletion in neutrophils or transferring of purified neutrophils from a knockout into a WT (or vice versa) or using a Klebsiella strain not reliant on neutrophils for control of infection), the authors need to change their conclusions to more accurately match the data shown throughout the manuscript.

The reviewers make an excellent point. We have not done those specific experiments. Therefore, this conclusion has been modified throughout the manuscript.

2) The flow cytometry data should also be presented as number of cells in a population (in addition to frequency of a population). For the flow cytometry data throughout the paper, the authors show % of live cells.

We thank the reviewers for the suggestion. All flow cytometry data are now presented as frequency and number of cells in a population.

3) It would be optimal if the authors had some data that shows a similar function for SKAP2 in human neutrophils. At the very least, text should be included that acknowledges the caveats (well known) of extrapolating data from mouse neutrophils to human neutrophils. The authors should be cautious in extrapolating their data to additional scenarios beyond those tested.

We agree with reviewers that such data would be great and is a key next step regarding whether there are differences between mouse and human immune response to *K. pneumoniae* infection, their ROS response, and the activation of various signal transduction pathways. We now discuss the caveats of using murine models to study *K. pneumoniae* in paragraph four of the Discussion, and cited specific references that are relevant to immune components investigated in our study. In addition, we have clarified that our data and proposed model is limited to murine studies throughout our manuscript.

Currently, to our knowledge, there are no commercially available inhibitors of Skap2 or its homologs. However, we note in the Introduction and Discussion sections that in addition to those associated with Crohn’s disease and Type I diabetes, there are other reported variants in Skap2 genes in humans in the Ensembl database (http://useast.ensembl.org/Homo_sapiens/Gene/Variation_Gene/Table?db=core;g=ENSG00000005020;r=7:26667068-26995239) although their functional phenotype is unknown. Thus, in the absence of other advances to genetically manipulate human hemopoietic stem cells and differentiate these into PMNs (or using HL-60 cells), future studies evaluating SKAP2 in human neutrophils will require isolating them from these individuals.

4) In Discussion paragraph four the authors mentioned published work that SKAP2-/- neutrophils exhibit a migration defect in response to sterile inflammation and imply there is no migration defect during infection. But the authors never show any data to support this implication. The authors assume that migration kinetics are the same because they measure the same number of neutrophils in the lungs during infection at specific timepoints, but they do not provide information on how many neutrophils were present in the blood and lungs of naïve mice, nor do they do a pulse labeling of newly differentiated neutrophils to track their rate of migration.

We agree. To provide more data about the levels of neutrophils in these mice, we have now included the frequency and number of neutrophils were present in the bone marrow naïve mice (Figure 1—figure supplement 1B-C), and cited a prior publication comparing the level of neutrophils present in the blood of naïve mice (Togni et al., 2005). In addition, we have clarified the level of neutrophils in PBS-treated lungs in Figure 1C and Figure 1—figure supplement 1A; levels of alveolar macrophages, dendritic cells, and resident monocytes from PBS-treated lungs are shown as dotted line in Figure 1—figure supplement 1E-J. We have also included preliminary data in the number of neutrophils recovered from the bronchoalveolar lavage of infected mice at 16 and 24 hours in Figure 1—figure supplement 1D. In all cases, there are no differences between the two strains of mice. However, as the reviewers point out these data do not prove that the migration kinetics are the same, and that a pulse labeling of newly differentiated neutrophils will be optimal in addressing this question. We have included this point in the Discussion.

5) In this same section, the authors mention that neutrophil survival could be affected by loss of SKAP2. This is a very important point in general and in particular for when analyzing their in vitro data. Did the authors observe a difference in survival of neutrophils from SKAP2 mice? How were each of their in vitro studies normalized for total live cells at each time point?

The viability of neutrophils collected for in vitro studies was assessed by trypan blue exclusion test during cell counting by hemocytometer immediately prior to aliquoting the neutrophils into assay conditions (subsection “Differentiating neutrophils from Hoxb8-immortalized stem cell lines”). The number of cells used per condition was based on the number of trypan blue excluded cells prior to functional studies. We have now included a sample of viability based on this assay for DIV neutrophils differentiated from 2 sets of Hoxb8 GMP (where GMP were harvested from different mice on different day) in Figure 3—figure supplement 1E. In some experiments, we also used Promega CellTiter-Glo Luminescent Cell viability assay to track neutrophil survival through the ATP measurement through the course of our functional studies; an example is shown in Figure 3—figure supplement 1F on unstimulated neutrophils, which was taken alongside a ROS assay.